# Large Vision-Language Models Get Lost in Attention

**Gongli Xi** [1 2]   **Ye Tian** [2 3]   **Mengyu Yang** [2 3]   **Huahui Yi** [4]   **Liang Lin** [5]   **Xiaoshuai Hao** [6]   **Kun Wang** [4]   **Wendong Wang** [2 3]

## Abstract

Despite the rapid evolution of training paradigms, the decoder backbone of large vision–language models (LVLMs) remains fundamentally rooted in the residual-connection Transformer architecture. Therefore, deciphering the distinct roles of internal modules is critical for understanding model mechanics and guiding architectural optimization. While prior statistical approaches have provided valuable attribution-based insights, they often lack a unified theoretical basis. To bridge this gap, we propose a unified framework grounded in *information theory and geometry* to quantify the **geometric and entropic nature** of residual updates. Applying this unified framework reveals a fundamental functional decoupling: **Attention acts as a subspace-preserving operator** focused on reconfiguration, whereas **FFNs serve as subspace-expanding operators** driving semantic innovation. Strikingly, further experiments demonstrate that replacing learned attention weights with predefined values (e.g., Gaussian noise) yields comparable or even superior performance across a majority of datasets relative to vanilla models. These results expose severe **misallocation and redundancy** in current mechanisms, suggesting that state-of-the-art LVLMs effectively "get lost in attention" rather than efficiently leveraging visual context. Our code is publicly available at this link.

[1]School of Cyberspace Security, Beijing University of Posts and Telecommunications, Beijing, China [2]State Key Laboratory of Networking and Switching Technology, Beijing University of Posts and Telecommunications, Beijing, China [3]School of Computer Science (National Pilot Software Engineering School), Beijing University of Posts and Telecommunications, Beijing, China [4]Nanyang Technological University, Singapore [5]Institute of Information Engineering, Chinese Academy of Sciences, Beijing, China [6]Xiaomi EV, Beijing, China. Correspondence to: Ye Tian <yetian@bupt.edu.cn>, Kun Wang <wang.kun@ntu.edu.sg>.

*Proceedings of the 43rd International Conference on Machine Learning*, Seoul, South Korea. PMLR 306, 2026. Copyright 2026 by the author(s).

## 1. Introduction

Large vision–language models (LVLMs) have rapidly evolved from large language models (LLMs) by extending Transformer-based sequence modeling to jointly process natural language and visual signals (Vaswani et al., 2017). Early vision–language representation learning (e.g., contrastive pretraining) established strong image–text alignment that later LVLMs could leverage as a visual grounding interface (Radford et al., 2021). Subsequent LVLMs increasingly unify pretrained vision encoders with LLM backbones, enabling few-shot multimodal generalization and instruction-following behavior at scale (Alayrac et al., 2022; Li et al., 2023a; Liu et al., 2023; Hao et al., 2025). In parallel, reasoning-oriented paradigms have further endowed these models with improved deliberation and problem-solving behaviors (Wei et al., 2022; Jaech et al., 2024; Guo et al., 2025; Zhang et al., 2025b; Tan et al., 2025). Despite the fast pace of architectural and training innovations, the dominant LVLM family remains fundamentally grounded in the Transformer architecture (Vaswani et al., 2017).

From an interpretability standpoint, the standard Transformer layer is composed of two core submodules, namely multi-head self-attention and feed-forward network (FFN), and each submodule is wrapped by residual connections, so that every submodule produces an additive update that is written back into a shared residual stream representation (Vaswani et al., 2017; Elhage et al., 2021; Skean et al., 2025). A common working *hypothesis* is that **attention blocks are the primary substrate for in-context reasoning**, implementing context-dependent algorithms such as induction/copy-based mechanisms (Olsson et al., 2022). In contrast, **FFNs are often characterized as storing and retrieving distributional associations**, behaving like key–value memories whose activated patterns can induce next-token distributions that resemble shallow n-gram continuations (Geva et al., 2021; Edelman et al., 2024).

To probe this modularity *hypothesis*, attention interpretability work has largely taken a **statistical perspective** that treats attention related signals as measurable proxies and attributes function via empirical distributions (Zhou et al., 2024; Kahardipraja et al., 2025), correlations (Jain & Wallace, 2019; Abnar & Zuidema, 2020), and controlled inter-

ventions (Serrano & Smith, 2019; Nam et al., 2025). More recently, this statistical toolkit has been extended to **visual attention** in LVLM decoders (Tian et al., 2026), where attention links text to visual tokens. Empirical analyses reveal systematic phenomena such as *visual attention sink* (Kang et al., 2025) and *visual attention drift* (Liu et al., 2025; Guan et al., 2026), which together indicate that models often under allocate attention to truly informative visual evidence. Given these advances, LVLM module-level interpretability still lacks a unifying **information theoretic and geometric** framework that can characterize, and explicitly contrast, how different submodules contribute to representation structure in multimodal settings. In contrast, the representation analysis literature for LLMs already uses such lenses to evaluate representation quality across depth (Razzhigaev et al., 2024; Wei et al., 2024) and to study joint dynamics (Skean et al., 2025; Tian et al., 2023). This gap motivates bringing these principled lenses into LVLM analysis to address the missing perspective and enable module specific and modality grounded comparisons.

To bridge this theoretical gap, we present a unified framework grounded in **information theory and differential geometry** to *quantify and contrast module-level functional contributions* in LVLM residual-stream computation. By adopting the manifold hypothesis (Bengio et al., 2013) for representation space, we introduce two complementary metrics: **Representation Information Discrepancy (RID)** and **Mixing Information Gain (MixIG)**. These metrics decompose the contribution of residual updates into two distinct geometric effects: *innovation*, which quantifies external information injection that expands the semantic subspace or alters spectral complexity, and *reconfiguration*, which measures the entropic redistribution of information within the existing support. We conduct extensive experiments across 15 state-of-the-art LVLMs spanning three dominant architectures on a broad suite of multimodal benchmarks. Our analysis reveals two profound insights: first, we quantitatively validate a sharp functional decoupling in Transformer residual stream computation: attention primarily performs entropic *reconfiguration* that preserves the existing representation support, whereas FFNs dominate *innovation* by introducing new semantic directions. Building on this division of labor, we further diagnose a systemic pathology in current LVLMs: decoder visual attention often fails to perform meaningful mixing over question-relevant visual evidence, and instead exhibits substantial redundancy, frequently getting lost in interaction patterns with limited contribution to informative updates.

Our main contributions are summarized as follows:

- **Theoretical Framework:** We propose a rigorous formalism based on the manifold hypothesis to define representational information. We introduce RID and MixIG

as dual metrics to quantify the geometric and entropic impact of residual updates, offering a generalized tool for probing representation dynamics.

- **Module-level Interpretability:** We provide a quantitative explanation of the distinct roles within Transformer blocks. We demonstrate that Attention and FFNs operate in orthogonal regimes—*reconfiguration* versus *innovation*—thereby substantiating the modularity hypothesis with geometric evidence.

- **Empirical Diagnostics:** We uncover critical inefficiencies in LVLM designs. Our results highlight that despite architectural scaling, current models suffer from severe informational redundancy in visual processing, suggesting that the integration of visual tokens is often computationally expensive yet informationally sparse.

## 2. Related work

**Interpretability of LLMs.** A large body of work studies what information is encoded in LLM representations and where it appears in the network (Belinkov & Glass, 2019). Early work uses lightweight linear probes on intermediate hidden states (Conneau et al., 2018; Hewitt & Manning, 2019; Belrose et al., 2023). Subsequent decoding based efforts, such as the tuned lens, map hidden states to vocabulary distributions (Belrose et al., 2023). Alongside probing and decoding, sparse feature learning approaches, including transcoders (Dunefsky et al., 2024) and sparse autoencoders (Cunningham et al., 2023), map representations into a sparse and more discrete feature space (Ameisen et al., 2025).

**Module Interpretability.** Module interpretability asks whether internal Transformer components provide meaningful explanations of model behavior. For attention, foundational studies show that raw attention weights can be an unreliable attribution signal (Jain & Wallace, 2019; Serrano & Smith, 2019; Wiegreffe & Pinter, 2019). To better capture how attention-mediated influence accumulates, attention rollout and attention flow estimate propagation across layers (Abnar & Zuidema, 2020; Kim et al., 2025). More recent work moves beyond token-level importance to head-level functionality by combining dataset-grounded attribution with causal validation (Nam et al., 2025; Kahardipraja et al., 2025; Zhou et al., 2024; Du et al., 2025). Complementarily, parameter-based approaches infer head functionality without per-prompt inference traces (Elhelo & Geva, 2025). In parallel, module-oriented analyses show that Feed-Forward layers can act as key–value memories (Geva et al., 2021; Qiu et al., 2024). By contrast, our work provides a unified information-theoretic and geometric framework that quantifies how different residual-stream updates contribute via innovation versus reconfiguration, enabling direct, module-wise comparison beyond attribution alone.

**Information theory in LLM interpretability.** Information-

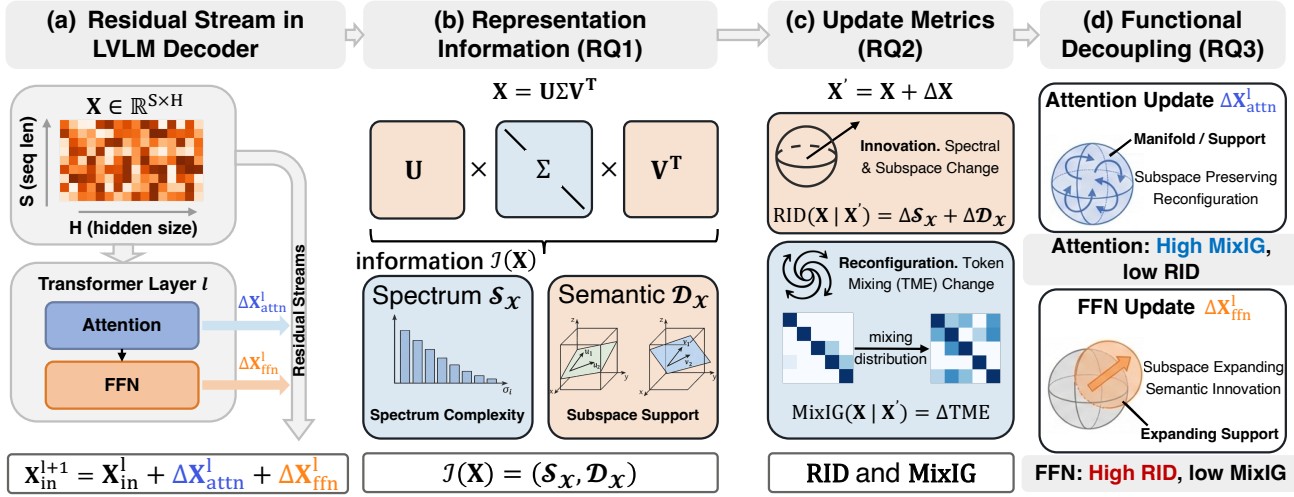

*Figure 1.* Overview of Our Interpretability Framework: **(a)** the LVLM residual stream; **(b)** representation information in $\mathbf{X}$, where SVD yields Spectrum $\mathcal{S}_\mathbf{X}$ and semantic support $\mathcal{D}_\mathbf{X}$; **(c)** update-level effects of $\Delta\mathbf{X}$, quantified by RID for innovation and MixIG for reconfiguration; **(d)** layer-wise functional decomposition, revealing an orthogonal division of labor where attention behaves as a subspace-preserving operator and FFNs act as subspace-expanding operators.

theoretic views frame interpretability in terms of information preservation, compression, and redundancy in representations. One line focuses on representation quality evaluation, using information and geometry motivated measures such as entropy, rank based quantities to assess whether embeddings preserve task relevant structure (Agrawal et al., 2022; Deb & Ogunfunmi, 2025; Li et al., 2025). A second line uses these measures for layerwise analyses, aiming to characterize how representational properties change across the network (Skean et al., 2025; Ali et al., 2025). A third line emphasizes compression and redundancy reduction as a model level capability that can correlate with performance and scaling trends (Wei et al., 2024; Yu et al., 2024; Havrilla & Liao, 2024). However, existing information theoretic work rarely provides *module-level interpretability* for module itself (Lai et al., 2021), especially in the LVLM setting.

Overall, we connect *module-level residual-stream updates* in LVLMs to information theory and geometry by *operationalizing* each update as an observable innovation–reconfiguration decomposition on representations. This framework turns prior statistically grounded module-level functional attributions into measurable information-flow statements, and it reveals that attention scores in current LVLMs contain substantial redundancy. Specifically, we replace part of the learned attention scores with random noise and find that model performance is largely preserved, even though this scoring step is a major computational bottleneck in standard self-attention, whose cost scales quadratically with sequence length.

## 3. A Unified Interpretability Framework for the Residual Stream

In this section, we first introduce the notation and research questions in Sec. 3.1. We then formalize representation information from an information-theoretic and geometric perspective in Sec. 3.2. Finally, in Sec. 3.3, we develop quantitative metrics for evaluating residual-stream updates.

### 3.1. Preliminaries

#### 3.1.1. MOTIVATION AND NOTATION

Consider an input $\mathcal{I}$, for example, a sequence of visual and language tokens. A multi-module neural network maps $\mathcal{I}$ to a hidden-state matrix $\mathbf{X} \in \mathbb{R}^{S \times H}$, where $S$ is the token length and $H$ is the hidden dimension. Throughout the forward pass, the representation is updated via residual connections. At each step, a module produces an additive update $\Delta\mathbf{X}$, yielding $\mathbf{X}_{\text{new}} = \mathbf{X}_{\text{old}} + \Delta\mathbf{X}$. This residual-update view raises three progressively refined questions:

**RQ1:** How should we quantify the information contained in a representation $\mathbf{X}$?

**RQ2:** How should we quantify what $\Delta\mathbf{X}$ contributes to $\mathbf{X}$?

**RQ3:** How can we use $\Delta\mathbf{X}$ to analyze and contrast the functional roles of different modules?

Answering these questions provides a principled foundation for tracking information flow across layers, characterizing when module updates are informative versus redundant, and understanding how different modules shape multimodal

representations during inference. For clarity, we summarize the notation used throughout the paper in *Appendix* Table 3.

### 3.1.2. RESIDUAL STREAM AND ATTENTION IN LVLMS

We next specify the LVLM setting and introduce the residual-stream view of Transformer decoding.

**Large Vision–Language Models (LVLMs).** We consider decoder-style LVLMs that process multimodal inputs by converting them into a single token sequence. Concretely, an image is encoded by a visual encoder and mapped through a modality projector into a sequence of visual tokens $\mathbf{X}^{(v)} \in \mathbb{R}^{S_v \times H}$. Textual inputs are tokenized into system and user tokens $\mathbf{X}^{(s)} \in \mathbb{R}^{S_s \times H}$ and $\mathbf{X}^{(q)} \in \mathbb{R}^{S_q \times H}$. We denote the concatenated input sequence by

$$\mathbf{X}^{(c)} = \big[\mathbf{X}^{(s)}, \mathbf{X}^{(v)}, \mathbf{X}^{(q)}\big] \in \mathbb{R}^{S_c \times H}, \quad S_c = S_s + S_v + S_q.$$

At decoding step $t$, the model generates an output token $y_t$ from

$$p(y_t \mid \mathbf{X}^{(c)}, \mathbf{y}_{<t}), \quad \mathbf{y}_{<t} = \{y_i\}_{i=1}^{t-1},$$

where $\mathbf{y}_{<t}$ determines the autoregressive context and $\mathbf{X}^{(c)}$ provides the multimodal conditioning.

**Attention in LVLM decoders.** Let the decoder have $L$ Transformer layers. At each layer $l$ and decoding step $t$, causal multi-head attention produces a normalized distribution over the *available* tokens, i.e., the concatenation of $S_c$ context tokens (system, visual, and question tokens) and the $(t-1)$ previously generated tokens. We denote the total attention domain size by $S_t = S_c + (t-1)$. The attention distribution at step $t$ is $\mathbf{a}_t^l \in [0,1]^{S_t}$ with $\sum_{i=1}^{S_t} a_{t,i}^l = 1$. Concretely, letting $\mathbf{q}_t^l \in \mathbb{R}^{d_k}$ be the query at step $t$ and $\mathbf{K}_t^l \in \mathbb{R}^{S_t \times d_k}$ be the key matrix formed from all available tokens up to step $t$ at layer $l$, we write

$$\mathbf{a}_t^l = \text{softmax}\Big(\frac{\mathbf{K}_t^l \mathbf{q}_t^l}{\sqrt{d_k}}\Big), \qquad \mathbf{a}_t^l \in [0,1]^{S_t},$$

which records, for each decoding step and layer, how the decoder allocates attention over *available* tokens.

**Residual Stream** Following the mathematical interpretation of the residual stream in Elhage et al. (2021), we view the layerwise hidden states as a residual stream that evolves via additive updates from each module. In our notation, the representation matrix at layer $l$ satisfies

$$\mathbf{X}_{\text{in}}^{l+1} = \mathbf{X}_{\text{in}}^l + \Delta\mathbf{X}_{\text{attn}}^l + \Delta\mathbf{X}_{\text{ffn}}^l, \qquad \mathbf{X}^l \in \mathbb{R}^{S \times H}.$$

### 3.1.3. THEORETICAL FOUNDATIONS

In this subsection, we introduce our foundational assumptions and the mathematical tools used to characterize a representation matrix $\mathbf{X} \in \mathbb{R}^{S \times H}$.

**Assumption 3.1** (Manifold hypothesis (Bengio et al., 2013))**.** Learned representations often concentrate near a low-dimensional structure embedded in a high-dimensional ambient space. This assumption motivates using low-rank spectral structure as a meaningful proxy for the effective degrees of freedom of $\mathbf{X}$. It also underpins a growing body of representation-centric studies in modern deep models (Wang et al., 2024a; Basile et al., 2024; Gardinazzi et al., 2025; Nishi et al., 2025).

**Definition 3.2** (Frobenius norm)**.** For $\mathbf{X} \in \mathbb{R}^{S \times H}$,

$$\|\mathbf{X}\|_F = \Big(\sum_{s=1}^S \sum_{h=1}^H \mathbf{X}_{s,h}^2\Big)^{\frac{1}{2}} = \sqrt{\text{tr}(\mathbf{X}^\top \mathbf{X})} = \Big(\sum_{i=1}^Q \sigma_i^2\Big)^{\frac{1}{2}}.$$

It measures the total energy of $\mathbf{X}$ in the ambient space.

## 3.2. Geometric Characterization of Representation Information on Matrix Manifolds (RQ1)

In what follows, we progressively answer the three research questions posed in Section 3.1.1. **RQ1** asks: *How should we quantify the information contained in a representation* $\mathbf{X}$*?* To quantify the information in $\mathbf{X}$, we adopt a geometric perspective based on the fixed-rank matrix manifold.

From differential geometry, the set of matrices with rank $r$,

$$\mathcal{M}_r = \{\mathbf{X} \in \mathbb{R}^{S \times H} : \text{rank}(\mathbf{X}) = r\},$$

admits a smooth Riemannian manifold structure (as an embedded submanifold in the ambient Euclidean space of matrices) (Vandereycken, 2013). For any $\mathbf{X} \in \mathcal{M}_r$, a compact singular value decomposition parameterizes $\mathbf{X}$ as $\mathbf{X} = \mathbf{U}\boldsymbol{\Sigma}\mathbf{V}^\top$ with $r$ positive singular values:

**Definition 3.3** (Singular Value Decomposition (Golub & Van Loan, 2013))**.** For any $\mathbf{X} \in \mathbb{R}^{S \times H}$, let $Q = \min\{S, H\}$. The SVD of $\mathbf{X}$ is

$$\mathbf{X} = \mathbf{U}\boldsymbol{\Sigma}\mathbf{V}^\top = \sum_{i=1}^Q \sigma_i \mathbf{u}_i \mathbf{v}_i^\top,$$

where $\mathbf{U} \in \mathbb{R}^{S \times Q}$ and $\mathbf{V} \in \mathbb{R}^{H \times Q}$ have orthonormal columns, $\boldsymbol{\Sigma} = \text{diag}(\sigma_1, \ldots, \sigma_Q)$ with $\sigma_1 \geq \cdots \geq \sigma_Q \geq 0$, and $(\mathbf{u}_i, \mathbf{v}_i)$ are the left and right singular vectors.

Under this parameterization, $\mathbf{X}$ is described by three geometric objects:

- **Left singular subspace** $\mathcal{C}(\mathbf{X}) = \text{span}(\mathbf{U}) \in \text{Gr}(r, S)$, capturing association structure in the token space;
- **Right singular subspace** $\mathcal{R}(\mathbf{X}) = \text{span}(\mathbf{V}) \in \text{Gr}(r, H)$, capturing semantic directions in the feature space;
- **Singular spectrum** $\boldsymbol{\Sigma} \in \mathbb{R}_+^r$, capturing the energy distribution across principal directions.

Here $\mathrm{Gr}(r, n)$ denotes the Grassmann manifold, the set of all $r$-dimensional linear subspaces of $\mathbb{R}^n$ (Absil et al., 2008).

Motivated by this geometry, we formalize the information contained in $\mathbf{X}$ as a pair

$$\mathcal{I}(\mathbf{X}) = (\mathcal{S}_{\mathbf{X}}, \mathcal{D}_{\mathbf{X}}).$$

Here $\mathcal{S}_{\mathbf{X}}$ denotes the *information complexity*, determined by the singular spectrum, and $\mathcal{D}_{\mathbf{X}}$ denotes the *information support*, determined by the left and right subspaces. We detail these two components next.

### 3.2.1. INFORMATION COMPLEXITY (SPECTRUM $\mathcal{S}_{\mathbf{X}}$)

Based on Theorem F.2 (Eckart & Young, 1936), the singular values determine the optimal rank-$k$ approximation error and therefore quantify how much of $\mathbf{X}$ can be captured by its leading principal directions. We thus summarize the concentration versus spread of the singular spectrum into an effective dimensionality using *effective rank* (eRank):

**Definition 3.4** (Rank and Effective rank (Roy & Vetterli, 2007))**.** For $\mathbf{X}$ with singular values $\{\sigma_i\}_{i=1}^Q$, the rank is

$$\mathrm{rank}(\mathbf{X}) = \big|\{i : \sigma_i > 0\}\big|.$$

Let $p_i = \sigma_i / \sum \sigma$ be the normalized singular spectrum. We define the Spectrum $\mathcal{S}_{\mathbf{X}}$ of the matrix as its effective rank:

$$\mathcal{S}_{\mathbf{X}} = \mathrm{eRank}(\mathbf{X}) = \exp\Big(-\sum_{i=1}^Q p_i \log p_i\Big).$$

This quantity corresponds to the *scale* component in the SVD-based representation, namely the singular spectrum $\boldsymbol{\Sigma}$.

### 3.2.2. INFORMATION SUPPORT (SUPPORT $\mathcal{D}_{\mathbf{X}}$)

This component corresponds to the Grassmann points $\mathcal{C}(\mathbf{X})$ and $\mathcal{R}(\mathbf{X})$ in the manifold parameterization. We view "semantics" as the linear subspaces occupied by the data in the ambient vector spaces; under the manifold hypothesis, high-dimensional semantic structure often concentrates near low-dimensional subspaces. Concretely, the column space $\mathcal{C}(\mathbf{X})$ (spanned by $\mathbf{U}$) specifies what semantic categories the layer representation can express, while the row space $\mathcal{R}(\mathbf{X})$ (spanned by $\mathbf{V}$) specifies linear dependency structure among tokens. In practice, we parameterize these Grassmann points using the orthonormal bases from SVD via the associated orthogonal projectors:

$$\mathbf{P}_{\mathcal{C}(\mathbf{X})} = \mathbf{U}\mathbf{U}^\top, \ \mathbf{P}_{\mathcal{R}(\mathbf{X})} = \mathbf{V}\mathbf{V}^\top, \ \mathcal{D}_{\mathbf{X}} = (\mathbf{P}_{\mathcal{C}(\mathbf{X})}, \mathbf{P}_{\mathcal{R}(\mathbf{X})})$$

which uniquely determine the supporting subspaces of $\mathbf{X}$.

**Discussion.** We have thus answered **RQ1** by formalizing the information contained in a representation $\mathbf{X}$ as two complementary components: the singular spectrum $\boldsymbol{\Sigma}$ encodes how energy is distributed across principal directions and thereby quantifies information complexity $\mathcal{S}_{\mathbf{X}}$, while the orthonormal factors $(\mathbf{U}, \mathbf{V})$ determine the supporting subspaces $\mathcal{C}(\mathbf{X}) = \mathrm{span}(\mathbf{U})$ and $\mathcal{R}(\mathbf{X}) = \mathrm{span}(\mathbf{V})$, fixing the geometric orientation of the representation in token and feature spaces and capturing structured semantics $\mathcal{D}_{\mathbf{X}}$.

### 3.3. Quantifying the Contribution of an Update $\Delta\mathbf{X}$ (RQ2)

In Section 3.2, we answered **RQ1** by defining the information in a representation as $\mathcal{I}(\mathbf{X}) = (\mathcal{S}_{\mathbf{X}}, \mathcal{D}_{\mathbf{X}})$. We now address **RQ2**: *How should we quantify what $\Delta\mathbf{X}$ contributes to $\mathbf{X}$?* Given an additive update $\mathbf{X}' = \mathbf{X} + \Delta\mathbf{X}$, its effect on $\mathbf{X}$ admits three complementary and collectively exhaustive categories under our decomposition:

1. **Spectrum change** (change in $\mathcal{S}_{\mathbf{X}}$): $\Delta\mathbf{X}$ reshapes the singular spectrum, inducing compression or expansion of the effective dimensionality, which reflects how information mass is redistributed across principal directions.
2. **Support change** (change in $\mathcal{D}_{\mathbf{X}}$): $\Delta\mathbf{X}$ perturbs the column and row subspaces, introducing or removing semantic support directions, namely a geometric shift in what the representation can express and how tokens linearly depend on one another.
3. **Internal interaction** (no external support): $\Delta\mathbf{X}$ remains within the existing support and acts by *reconfiguration*, namely reorganizing and reallocating information already present in $\mathbf{X}$ without injecting new support directions.

The first two categories reflect external information injection that changes complexity or support. The third captures *reconfiguration*, since it reflects internal redistribution within the existing information support. We next define measures for external information injection and reconfiguration.

### 3.3.1. MEASURING EXTERNAL INFORMATION INJECTION

**Spectrum change.** We quantify the spectrum change by the eRank variation, normalized to lie in $[0, 1]$.

$$\Delta\mathcal{S}(\mathbf{X} \mid \mathbf{X}') = \frac{\big|\mathrm{eRank}(\mathbf{X}') - \mathrm{eRank}(\mathbf{X})\big|}{\min\{S, H\}}.$$

**Support innovation.** To measure how much new support is introduced by $\Delta\mathbf{X}$, we use the innovation view from least squares, where innovation is the residual after projecting onto a reference subspace:

**Definition 3.5** (Subspace Innovation)**.** Let $\mathcal{U} \subseteq \mathbb{R}^d$ be a linear subspace with orthogonal projector $\mathbf{P}_{\mathcal{U}}$. For an observation $\mathbf{y} \in \mathbb{R}^d$, the least-squares prediction in $\mathcal{U}$ is $\hat{\mathbf{y}} = \mathbf{P}_{\mathcal{U}}\mathbf{y}$. The innovation is the residual (Hassibi et al., 2000)

$$\tilde{\mathbf{y}} = \mathbf{y} - \hat{\mathbf{y}} = (\mathbf{I} - \mathbf{P}_{\mathcal{U}})\mathbf{y}.$$

Analogously, we define the *support innovation* of the update $\Delta \mathbf{X}$ relative to $\mathbf{X}$ as the energy that lies in the orthogonal complements of the column and row spaces of $\mathbf{X}$. Let $\mathbf{P}_{\mathcal{C}(\mathbf{X})}$ and $\mathbf{P}_{\mathcal{R}(\mathbf{X})}$ be the orthogonal projectors onto $\mathcal{C}(\mathbf{X})$ and $\mathcal{R}(\mathbf{X})$. We define

$$\Delta \mathcal{D}(\mathbf{X} \mid \mathbf{X}') = \frac{\left\| (\mathbf{I} - \mathbf{P}_{\mathcal{C}(\mathbf{X})}) \mathbf{X}' \right\|_F + \left\| \mathbf{X}'(\mathbf{I} - \mathbf{P}_{\mathcal{R}(\mathbf{X})}) \right\|_F}{2 \times \|\mathbf{X}'\|_F}.$$

**Two-dimensional innovation vector.** The two terms above capture complementary channels of external information injection. Spectrum change $\Delta \mathcal{S}$ measures variation in effective dimensionality, while support innovation $\Delta \mathcal{D}$ measures novelty in the column and row subspaces. We therefore first represent innovation as a two-dimensional quantity:

$$\Delta \mathcal{I}(\mathbf{X} \mid \mathbf{X}') = \big( \Delta \mathcal{S}(\mathbf{X} \mid \mathbf{X}'), \Delta \mathcal{D}(\mathbf{X} \mid \mathbf{X}') \big).$$

Using either component alone may miss complementary cases, such as subspace change with little spectral variation. Since both components are normalized to comparable ranges, we aggregate them into a scalar summary score, defined next.

**Definition 3.6** (**Representation Information Discrepancy (RID)**). Given two representation matrices $\mathbf{X}, \mathbf{X}' \in \mathbb{R}^{S \times H}$, we define the *Representation Information Discrepancy* as the sum of the spectrum change and the support innovation:

$$\text{RID}(\mathbf{X} \mid \mathbf{X}') = \Delta \mathcal{S}(\mathbf{X} \mid \mathbf{X}') + \Delta \mathcal{D}(\mathbf{X} \mid \mathbf{X}').$$

RID measures how a representation changes in spectral complexity and subspace novelty, and satisfies $\text{RID} \in [0, 2]$ (Lemma F.1). Since positional encoding and parameterization effects make RID rarely exactly zero in practice, we introduce a tolerance $\epsilon > 0$ and treat $\mathbf{X}'$ as information-preserving relative to $\mathbf{X}$ whenever $\text{RID}(\mathbf{X} \mid \mathbf{X}') \approx \epsilon$; concretely, we set $\epsilon_{\text{RoPE}} = \text{RID}\big( \mathbf{X}_{\text{in}}^{(\text{RoPE})} \mid \mathbf{X}_{\text{in}}^{(\text{no-RoPE})} \big)$, which calibrates $\epsilon$ to the intrinsic discrepancy induced by Rotary Positional Encoding (RoPE) (Su et al., 2024).

### 3.3.2. MEASURING RECONFIGURATION

Another effect of $\Delta \mathbf{X}$ is *reconfiguration*, namely redistributing information within the existing support. We measure this internal redistribution via a token-to-token mixing entropy.

**Definition 3.7** (Token Mixing Entropy (TME)). Given a hidden-state matrix $\mathbf{X} \in \mathbb{R}^{S \times H}$ with row vectors $\mathbf{x}_t \in \mathbb{R}^H$, define $\tilde{\mathbf{x}}_t = \mathbf{x}_t / \|\mathbf{x}_t\|_2$ as the unit direction vector. We form a token-to-token mixing distribution by mapping pairwise token similarities to $[0, 1]$ and then row-normalizing

$$P_{t,j} = \frac{\frac{\tilde{\mathbf{x}}_t^\top \tilde{\mathbf{x}}_j + 1}{2}}{\sum_{k=1}^{S} \frac{\tilde{\mathbf{x}}_t^\top \tilde{\mathbf{x}}_k + 1}{2}}, \qquad t, j \in \{1, \ldots, S\}.$$

The Token Mixing Entropy is the average Shannon entropy of these distributions:

$$\text{TME}(\mathbf{X}) = -\frac{1}{S} \sum_{t=1}^{S} \sum_{j=1}^{S} P_{t,j} \log P_{t,j}.$$

$\text{TME}(\mathbf{X})$ provides an operational measure of token-level interaction by summarizing how broadly each token mixes with the rest of the sequence. It constructs a token-to-token mixing distribution from pairwise similarity and quantifies its uncertainty via entropy, so **larger** TME indicates more diffuse, globally shared interactions, whereas smaller TME indicates more concentrated, selective mixing.

**Definition 3.8** (**Mixing Information Gain (MixIG)**). For an updated representation $\mathbf{X}' = \mathbf{X} + \Delta \mathbf{X}$, we define the mixing information gain as the change in token mixing entropy:

$$\text{MixIG}(\mathbf{X} \mid \mathbf{X}') = \text{TME}(\mathbf{X}') - \text{TME}(\mathbf{X}).$$

This quantity captures how strongly the update increases or decreases token-to-token mixing, and thus serves as an operational measure of *reconfiguration* within the existing information support.

**Discussion.** In this section, we answer **RQ2** with two complementary metrics: RID and MIxIG. RID quantifies *innovation* by measuring how $\Delta \mathbf{X}$ changes the representation through spectral complexity shifts and support novelty, indicating external information injection beyond the current subspace. MIxIG quantifies *reconfiguration* by measuring how $\Delta \mathbf{X}$ reshapes token to token mixing within the existing support, capturing internal redistribution of information without introducing new support directions.

## 4. Redundancy and Misallocation in LVLM Visual Attention (RQ3)

In this section, we build on our theoretical framework to answer **RQ3**: *How can we use $\Delta \mathbf{X}$ to analyze and contrast the functional roles of different modules?* Through experiments, we uncover a common pathology in Transformer-based LVLMs: **models can get lost in attention**. We first describe the experimental setups in Section 4.1. Then, in Section 4.2, we use RID and MIxIG to show that different modules exhibit orthogonal functional roles, complementing prior statistically grounded interpretability studies (Kang et al., 2025; Geva et al., 2021). Finally, in Section 4.3, we replace attention scores with predefined values, and the results indicate substantial redundancy in existing LVLM attention.

### 4.1. Experimental Setups

**Model settings.** We evaluate 15 open-source LVLM variants spanning three mainstream architectures. Specifically, we consider Qwen-family models (Qwen-2.5-VL

(Team, 2025), CoF (Wei et al., 2022), Reverse (Wu et al., 2025), MM-Eureka (Meng et al., 2025), Orsta (Ma et al., 2025b), Ocean-R1 (Ming et al., 2025)), LLaVA-1.5-family models (LLaVA-1.5 (Liu et al., 2024a), Yi-VL (AI et al., 2024)), and LLaVA-NeXT-family models (LLaVA-OneVision (Li et al., 2024b), Mistral-1.6 and Vicuna-1.6 (Liu et al., 2024b)).

**Tasks and benchmarks.** Our experiments are conducted on a broad suite of multimodal benchmarks, including POPE (Li et al., 2023b), 3DSRBench (Ma et al., 2025a), Real-WorldQA (Visheratin, 2024), MMMU (Yue et al., 2023), VMC-Bench (Zhang et al., 2025c), MathVista (Lu et al., 2023), and HallusionBench (Guan et al., 2024). Together, these benchmarks evaluate LVLM capabilities from basic visual perception to advanced multimodal reasoning. Details on the benchmarks are provided in the Appendix C.1.

### 4.2. Interpreting the Functional Roles of Attention and FFN

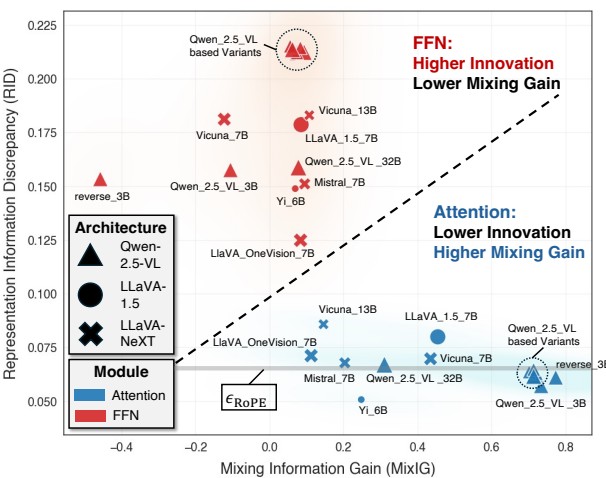

*Figure 2.* Model-wise RID and MixIG for Attention and FFN. Across architectures and training variants, a clear and consistent separation emerges between attention and FFN contributions, indicating that our framework captures an intrinsic functional distinction between the two submodules. Specifically, $\epsilon_{\text{RoPE}} = 0.062$.

To systematically dissect the information dynamics within the residual stream, we track the evolution of RID and MixIG across all layers $l$, using a random sample of 1000 instances from each dataset. We design three comparative settings to isolate the contributions of learned architectural components versus stochastic interference:

1. **Stochastic Baselines ($\mathbf{X}_{\text{noise}}^l$):** We introduce two randomization strategies to validate metric sensitivity and isolate learned functional properties: (1) **Noise $\Delta$**, where the attention update is replaced by Gaussian noise match-

ing the empirical moments of $\Delta\mathbf{X}_{\text{attn}}$, serving as a negative control to verify the detection of unstructured, off-manifold perturbations; (2) **Noise QKV**, where learned weight matrices are replaced by Gaussian initializations, serving to demonstrate that the subspace-preserving nature of attention is a learned behavior, as unoptimized linear transformations would otherwise significantly perturb the feature space. In both cases, we match the noise mean to that of $\Delta\mathbf{X}_{\text{attn}}$ (Theorem F.3).

2. **Attention Contribution:** We measure the transition from input to post-attention states via $\text{RID}(\mathbf{X}_{\text{in}}^l \mid \mathbf{X}_{\text{attn}}^l)$ and $\text{MixIG}(\mathbf{X}_{\text{in}}^l \mid \mathbf{X}_{\text{attn}}^l)$.

3. **FFN Contribution:** We measure the transition from post-attention to post-FFN states via $\text{RID}(\mathbf{X}_{\text{attn}}^l \mid \mathbf{X}_{\text{ffn}}^l)$ and $\text{MixIG}(\mathbf{X}_{\text{attn}}^l \mid \mathbf{X}_{\text{ffn}}^l)$.

The aggregated statistics are shown in Table 1 and Figure 2, while layer-wise trajectories are illustrated in Figure 3.

*Table 1.* Module-wise RID and MIxIG with qualitative signatures.

| Module | RID | MixIG | Characteristic |
|---|---|---|---|
| Noise $\Delta$ | 0.61 | -0.80 | Very high RID |
| Noise **QKV** | 0.44 | -0.50 | Negative MixIG |
| Attention | 0.06 | **0.61** | Low RID, High MixIG |
| Feed-Forward | **0.21** | 0.02 | High RID, Low MixIG |

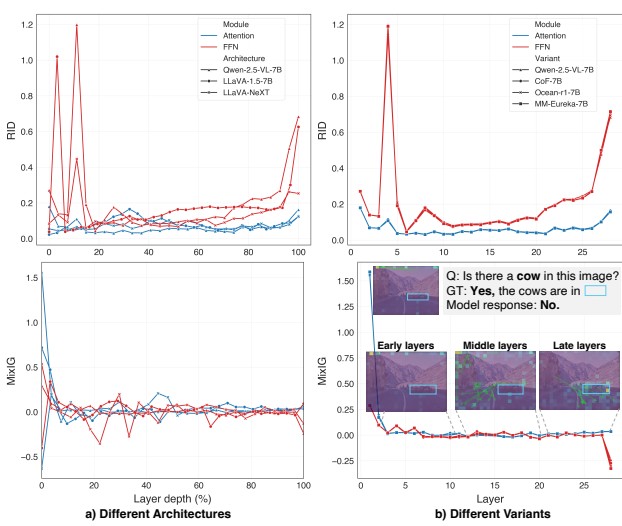

*Figure 3.* Layer-wise RID and MixIG for Attention and FFN. More sample visualizations are provided in the Figures 5–10.

Our observations are as follows:

**Obs ❶. Metric Discriminability and Subspace Sensitivity.** Table 1 validates our metrics through stochastic baselines. Noise $\Delta$ and Noise **QKV** serve as negative controls for testing whether RID and MIxIG can distinguish structured

*Table 2.* Benchmark results under different SAP modes. We **bold** the best results and underline the runner-ups *within each model*.

| Model / Variant | Affected Layers | POPE | RWQA | 3dSRBench | MMMU | VMCBench | HallusionBench | MathVista |
|---|---|---|---|---|---|---|---|---|
| `Qwen-2.5-VL-3B` | / | 86.13 | 59.35 | 53.46 | 47.78 | 72.31 | 66.97 | 61.5 |
|   + Vis. Attn. | | **87.58** | 61.38 | 53.94 | **48.29** | **72.67** | 68.66 | 61.6 |
|   + Patch Comp. | [1, 27] | 87.47 | **61.62** | **54.14** | 47.88 | 72.59 | **69.19** | **61.7** |
|   + Noise | | 87.40 | 60.52 | 53.85 | **48.29** | 72.66 | 69.09 | 61.6 |
| `Qwen-2.5-VL-7B` | / | 86.54 | 65.75 | 55.63 | **51.77** | 74.34 | 69.19 | **63.3** |
|   + Vis. Attn. | | 87.62 | 66.14 | 56.60 | 51.18 | 74.77 | 70.98 | 63.1 |
|   + Patch Comp. | [1, 27] | **87.73** | **66.54** | **56.74** | 51.32 | **74.80** | **71.40** | 63.1 |
|   + Noise | | 87.51 | **66.54** | 56.56 | 51.76 | 74.76 | 70.35 | 62.9 |
| `LLaVA-1.5-7B` | / | 74.38 | 47.71 | 47.53 | 34.12 | 48.71 | 41.63 | 21.9 |
|   + Vis. Attn. | | **75.79** | 50.20 | 48.65 | 34.71 | 52.23 | **44.29** | 23.2 |
|   + Patch Comp. | [18, 23] | 75.30 | **50.85** | **48.96** | 35.18 | **52.29** | 42.42 | **23.6** |
|   + Noise | | 75.02 | 47.58 | 48.81 | **35.23** | 50.70 | 42.69 | 22.9 |
| `LLaVA-OneVision-7B` | / | 86.21 | 56.73 | 55.54 | 41.51 | 66.79 | 46.94 | 63.7 |
|   + Patch Comp. | [21, 27] | **87.78** | **60.26** | **57.22** | **42.76** | **68.79** | **47.48** | 63.7 |
|   + Noise | | 87.28 | 59.09 | 56.72 | 40.99 | 67.80 | 47.03 | **64.3** |

module updates from unstructured perturbations. The substantially higher RID and negative MixIG of both baselines show that unstructured perturbations are correctly identified as off-subspace disruptions with reduced token mixing, confirming that the low-RID, positive-MixIG profile of attention reflects a learned structured update rather than a metric artifact.

**Obs ❷. The Orthogonal Roles of Attention and FFN.** Figure 2 shows a consistent separation between attention and FFN across 15 LVLM variants. Attention updates exhibit negligible innovation (on the order of $\epsilon_{\text{RoPE}}$) but strong reconfiguration, acting as a *subspace-preserving operator*. In contrast, FFN updates exhibit substantial innovation with weak reconfiguration, acting as a *subspace-expanding operator*. Together, these results quantify a clear division of labor: attention primarily *contextualizes* existing information via rearrangement, whereas FFNs primarily *compute* new semantic features via subspace expansion.

**Obs ❸. Misallocation in visual attention.** The layer-wise analysis in Figure 3 suggests a heterogeneous role of attention across depth: while some layers exhibit pronounced reconfiguration (e.g., Layer 0 and layers around $40\%$ depth), cross-token interactions remain sparse in most layers. Motivated by this pattern, we further visualize attention-mediated cross-patch interactions in Figure 3(b) by linking patch pairs whose query–key score $\geq 0.1$. We model patch interactions as a graph and measure the degree share of question-relevant regions: this share is substantially lower for incorrect samples ($4.2\%$) than for correct ones ($13.1\%$), exposing a systematic *misallocation* of visual attention in current LVLM decoders. We further discuss this analysis in Appendix E.

**Summary.** In this section, we validate the discriminability of our metrics (**Obs ❶**) and confirm a robust module-level functional separation across diverse LVLM variants (**Obs ❷**). We further find that attention often fails to allocate and reorganize information around question-relevant visual evidence (**Obs ❸**). This naturally raises a follow-up question: *if attention scores exhibit such misallocation, are they largely redundant and replaceable?* We answer this question in the next section via targeted interventions.

## 4.3. Replacing Attention Scores with Priors

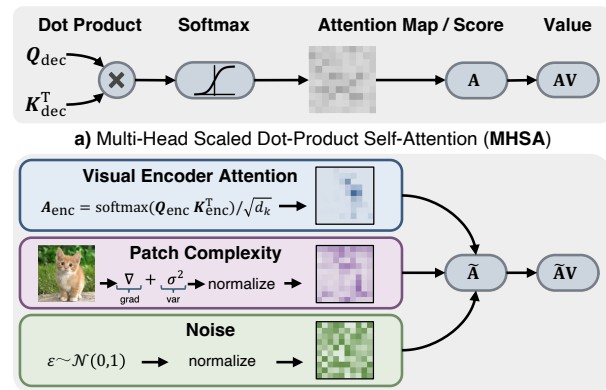

**a)** Multi-Head Scaled Dot-Product Self-Attention (**MHSA**)

**b)** MHSA Replacement with Shared Attention Prior (**SAP**)

*Figure 4.* MHSA Replacement with Shared Attention Prior. **Causal masking is still applied after the replacement.**

To further validate that a substantial portion of LVLM attention computation is redundant, we intervene on the decoder by replacing attention scores in selected layers with shared attention prior (SAP). As illustrated in Figure 4, we consider three replacement modes: *(i) Visual-encoder attention*, which injects attention maps derived from the visual encoder; *(ii) Patch complexity*, which uses a precomputed patch-wise complexity prior based on within-patch color variance and edge-gradient magnitude; and *(iii) Noise*, which substitutes scores with Gaussian noise. **Details of**

**SAP experiments** are provided in Appendix C.3.

Table 2 reports the SAP replacement results on three backbone families (`Qwen-2.5-VL`, `LLaVA-1.5`, and `LLaVA-NeXT`). **Detailed ablations on affected layers and heads**, as well as experiments on larger models and more variants, are provided in Appendix D.

**Obs ❹. Substantial redundancy in LVLM visual attention.** Across models and benchmarks (Table 2), replacing decoder attention scores with these predefined patterns does not degrade performance and can even yield improvements. This indicates that, for current LVLMs, a large fraction of visual-attention scoring is not functionally necessary, revealing substantial redundancy in decoder visual attention. This observation is consistent with recent visual token pruning works (Wen et al., 2025; Zhang et al., 2025a).

## 5. Discussion and Conclusion

We propose a unified theoretical framework for assessing how residual-stream updates shape representations in large models. Applying it to LVLMs reveals a consistent module-level functional separation, where attention primarily supports token-level reconfiguration while FFNs drive innovation, and further diagnoses a pervasive failure mode in current decoders: visual attention often misallocates interaction away from question-relevant evidence. Motivated by this deficiency, we conduct a **proof-of-concept** intervention by replacing attention scores in selected layers with simple predefined priors, and observe little to no degradation in capability, suggesting substantial redundancy in learned scoring. Beyond these specific findings, our framework and empirical protocol offer a general tool for evaluating residual-update mechanisms across model families and motivate targeted attention-centric optimization.

In conclusion, our framework turns LVLM residual updates into measurable innovation–reconfiguration dynamics and provides evidence that current Transformer-based LVLMs can *get lost in attention*. Future work includes extending the analysis to training-time dynamics and leveraging the observed redundancy to design more efficient attention mechanisms or regularizers that preserve useful mixing while reducing unnecessary scoring.

## Impact Statement

This paper presents work whose goal is to advance the field of Large Vision–Language Model Interpretability. There are many potential societal consequences of our work, none which we feel must be specifically highlighted here.

## Acknowledgments

This work is supported by the Beijing Nova Program under Grant 2023140.

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

## A. Notations

We summarize the notation used throughout this paper in Table 3.

*Table 3.* Notations.

| Notation | Description |
| --- | --- |
| $\mathbf{X} \in \mathbb{R}^{S \times H}$ | Hidden-state / residual-stream representation matrix with token length $S$ and hidden size $H$ |
| $\mathbf{X}_{\text{new}}, \mathbf{X}_{\text{old}}, \Delta\mathbf{X}$ | Updated representation, pre-update representation, and the additive residual update, $\mathbf{X}_{\text{new}} = \mathbf{X}_{\text{old}} + \Delta\mathbf{X}$ |
| $\mathbf{X}_{\text{in}}^l, \mathbf{X}_{\text{attn}}^l, \mathbf{X}_{\text{ffn}}^l$ | Layer-$l$ residual-stream states: layer input, post-attention state, and post-FFN state |
| $\Delta\mathbf{X}_{\text{attn}}^l, \Delta\mathbf{X}_{\text{ffn}}^l$ | Module-wise residual updates at layer $l$: $\Delta\mathbf{X}_{\text{attn}}^l = \mathbf{X}_{\text{attn}}^l - \mathbf{X}_{\text{in}}^l, \Delta\mathbf{X}_{\text{ffn}}^l = \mathbf{X}_{\text{ffn}}^l - \mathbf{X}_{\text{attn}}^l$ |
| $\mathbf{X} = \mathbf{U}\mathbf{\Sigma}\mathbf{V}^\top$ | Singular value decomposition (SVD) of $\mathbf{X}$ with orthonormal factors $\mathbf{U}, \mathbf{V}$ and singular spectrum $\mathbf{\Sigma}$ |
| $\mathcal{I}(\mathbf{X}) = (\mathcal{S}_{\mathbf{X}}, \mathcal{D}_{\mathbf{X}})$ | Representation information, decomposed into spectrum complexity $\mathcal{S}_{\mathbf{X}}$ and support $\mathcal{D}_{\mathbf{X}}$ |
| $\mathcal{C}(\mathbf{X}), \mathcal{R}(\mathbf{X})$ | Column space and row space of $\mathbf{X}$ (Grassmann points) |
| $\text{span}(\mathbf{U}), \text{span}(\mathbf{V})$ | Left and right singular subspaces induced by SVD factors $\mathbf{U}$ and $\mathbf{V}$ |
| $\mathbf{P}_{\mathcal{U}}$ | Orthogonal projector onto a subspace $\mathcal{U}$ |
| $\Delta\mathcal{S}(\mathbf{X} \mid \mathbf{X}'), \Delta\mathcal{D}(\mathbf{X} \mid \mathbf{X}')$ | Spectrum change and support innovation when transitioning from $\mathbf{X}$ to $\mathbf{X}'$ |
| $\text{RID}(\mathbf{X} \mid \mathbf{X}')$ | Representation Information Discrepancy, measuring update-induced innovation via spectrum change plus support innovation |
| $\text{TME}(\mathbf{X})$ | Token Mixing Entropy, an entropy-based measure of token-to-token mixing in $\mathbf{X}$ |
| $\text{MixIG}(\mathbf{X} \mid \mathbf{X}')$ | Mixing Information Gain, defined as $\text{TME}(\mathbf{X}') - \text{TME}(\mathbf{X})$ to quantify reconfiguration |
| $\mathbf{Q}, \mathbf{K}, \mathbf{V}, \mathbf{A}$ | Query, key, value, and attention weights (attention distribution / matrix) |
| $\mathbf{X}_{\text{in}}^{\text{(rope)}}, \mathbf{X}_{\text{in}}^{\text{(no-rope)}}$ | Layer-input representations with RoPE positional encoding enabled vs. disabled (for calibrating intrinsic discrepancy) |

## B. Comparison with Prior Work

As multi-modal models are increasingly deployed across diverse applications (Liang et al., 2026a;b; Feng et al.; 2026; Feng & Ge, 2025), concerns around reliability, privacy, and auditing have also become increasingly important (Abadi et al., 2016; Papernot et al., 2017; Xu et al., 2025), making it necessary to explain their internal mechanisms. Prior module-level interpretability studies have largely relied on attribution, tracing, or component-specific functional analyses. For attention, this line examines whether attention weights faithfully explain predictions, how attention-mediated influence propagates across layers, or which heads implement specific functions (Jain & Wallace, 2019; Serrano & Smith, 2019; Wiegreffe & Pinter, 2019; Abnar & Zuidema, 2020). For FFNs, prior work shows that feed-forward layers can behave as key–value memories that associate textual patterns with output distributions (Geva et al., 2021). These approaches are valuable for localizing where a behavior or stored pattern appears. In contrast, our framework asks a different question: how does each module transform the shared residual stream? We therefore characterize updates at the representation level through innovation and reconfiguration, rather than assigning a behavior to a specific token, head, neuron, or memory slot.

This difference also changes the diagnostic perspective. Prior methods are often strongest at identifying what function is present in a model, such as token attribution, head functionality, or stored associations. For example, causal tracing and editing methods localize factual associations in feed-forward modules (Meng et al., 2022; 2023; Fang et al., 2025), circuit analyses identify knowledge-related pathways (Yao et al., 2024), and head-level studies characterize which attention heads matter for in-context learning (Yin & Steinhardt, 2025). In multimodal settings, related work further studies where visual and textual information is stored and transferred across MLLM components (Basu et al., 2024), while FFN analyses examine how feed-forward blocks reshape contextualization patterns (Kobayashi et al., 2024). Our framework instead diagnoses what is insufficient or excessive in a residual update itself. RID asks whether a module injects new representational structure through spectral or subspace change, while MixIG asks whether a module meaningfully redistributes token-level information. Thus, innovation and reconfiguration are not direct substitutes for memory, retrieval, or attribution; they are update-level properties of the residual stream. This makes the analysis actionable, because weak innovation or weak reconfiguration can be directly linked to a module, layer, or intervention target.

The resulting conclusions are therefore complementary to prior work rather than redundant with it. For FFNs, memory-based interpretations explain how parameters can store and retrieve patterns, whereas our analysis measures how the FFN update changes representation geometry regardless of whether the source is parametric memory or contextual computation. For attention, circuit-level studies explain what algorithms attention can implement, whereas our claim concerns the visual side of current LVLM decoders: many attention updates show limited useful visual reconfiguration, and their score computation can often be replaced by simple priors without harming performance. In this sense, our work shifts the focus from identifying existing functions to diagnosing residual-stream deficiencies, revealing that current LVLMs do not consistently convert expensive visual attention scoring into necessary output-discriminative information flow.

# C. Details

## C.1. Dataset Details

Our experiments are conducted on a suite of benchmarks that probe complementary capabilities, spanning basic visual perception through advanced multimodal reasoning and robustness, including 3D and spatial reasoning, real-world question answering, multidisciplinary knowledge, general-purpose multimodal understanding, mathematical reasoning, and hallucination-related robustness. Detailed descriptions are provided below.

**POPE** (Li et al., 2023b). POPE is a diagnostic benchmark for *object hallucination* in LVLMs, it contains **9,000** questions split into three complementary subsets (random, popular, adversarial) to stress different hallucination modes. We conduct the experiments in Section 4.2 on POPE.

**3DSRBench** (Ma et al., 2025a). 3DSRBench targets *3D and spatial reasoning* by evaluating whether a model can infer geometric relations beyond surface-level recognition. It includes **1,500** visual QA problems spanning diverse 3D reasoning skills (e.g., relative depth, viewpoint-dependent relations, and compositional spatial constraints). The dataset is intended to separate "seeing" from "reasoning in 3D space" under multimodal inputs.

**RealWorldQA** (Visheratin, 2024). RealWorldQA evaluates *real-world visual question answering* on everyday imagery, emphasizing practical robustness rather than curated or synthetic settings. It contains **765** real-world images paired with questions, covering varied scenes and conditions that commonly challenge LVLM grounding.

**MMMU** (Yue et al., 2023). MMMU is a large-scale benchmark for *multidisciplinary multimodal understanding and reasoning*, spanning many academic domains. It contains **11,500+** questions across **30** subjects, covering both knowledge-intensive understanding and higher-level reasoning with visual inputs. Because evaluation on the full test set is restricted, we follow the widely adopted protocol in prior work and conduct our experiments on the `validation` split (900 samples).

**VMC-Bench** (Zhang et al., 2025c). VMC-Bench evaluates *general multimodal understanding* with an emphasis on challenging, automatically constructed multiple-choice questions. It transforms 20 widely-used VQA datasets into a unified multiple-choice benchmark. These datasets can be broadly categorized to assess general capabilities of VLMs (VQAv2 (Goyal et al., 2017), OKVQA (Marino et al., 2019), MMVet (Yu et al., 2023), VizWiz (Gurari et al., 2018), A-OKVQA (Schwenk et al., 2022), MMStar (Chen et al., 2024), SEEDBench (Li et al., 2024a)), reasoning capabilities (MathVision (Wang et al., 2024b), GQA (Hudson & Manning, 2019), MMMU (Yue et al., 2023), RealWorldQA (Visheratin, 2024), MathVista (Lu et al., 2023), ScienceQA (Lu et al., 2022)), OCR tasks (OCRVQA (Mishra et al., 2019), TextVQA (Singh et al., 2019)), and document and chart understanding (DocVQA (Mathew et al., 2021), InfoVQA (Mathew et al., 2022), ChartQA (Masry et al., 2022), TableVQABench (Kim et al., 2024), AI2D (Kembhavi et al., 2016)).

VMC-Bench contains **9,018** questions and is used to stress-test model discrimination among closely competing options.

**MathVista** (Lu et al., 2023). MathVista focuses on *visual mathematical reasoning*, requiring models to combine perception (reading diagrams, charts, or scenes) with mathematical problem solving. It contains **5,141** QA instances covering a wide range of math-reasoning skills grounded in visual context. Because the official MathVista test evaluation is not publicly available, we conduct our experiments on the `testmini` split (1,000 samples).

**HallusionBench** (Guan et al., 2024). HallusionBench is a targeted benchmark for *hallucination-related robustness*, separating failures caused by visual misperception (illusion-like cases) from those caused by language priors. It contains **1,129** image–question pairs constructed to systematically elicit hallucination behaviors under controlled conditions.

## C.2. Experimental Details

**Dataset settings.** For each benchmark, we follow a consistent evaluation protocol across all models. Specifically, we feed every image–question pair from the dataset to the model under the same input formatting and inference configuration, and compute the corresponding task metric using the official evaluation script whenever available.

**Model settings.** Within each model category, we adopt a unified inference setup to ensure fair comparison. We group the evaluated LVLMs into three categories.

**(i) General-purpose LVLMs.** This category includes `Qwen-2.5-VL`, `LLaVA-1.5`, `Yi`, `LLaVA-OneVision`, `Mistral-1.6`, and `Vicuna-1.6`. For these models, we directly input the dataset image–question pair using their default chat templates.

**(ii) Vision-query optimized LVLMs.** This category includes `Reverse` and `CoF`. For these models, we follow the inference and prompting settings specified in their respective papers to reproduce their intended evaluation protocol.

**(iii) Reasoning-oriented LVLMs.** This category includes `MM-Eureka`, `Orsta`, and `Ocean-R1`. For these models, we append an explicit reasoning trigger to encourage open-ended deliberation, and extract the final prediction from the `<answer>` tags in the generated output.

> **Reasoning-trigger prompt**
>
> You FIRST think about the reasoning process as an internal monologue and then provide the final answer. The reasoning process MUST BE enclosed within `<think>` `</think>` tags. The final answer MUST BE in `<answer>` `</answer>` tags.

**Generation hyperparameters.** We use the following decoding parameters for all experiments, and keep all unspecified options at their default values:

$$\texttt{max\_new\_tokens} = 1024, \quad \texttt{output\_attentions} = \texttt{True}, \quad \texttt{return\_dict\_in\_generate} = \texttt{True}.$$

**Evaluation details.** We follow the official evaluation protocols of each dataset and report *accuracy* as the primary metric. For open-ended outputs (e.g., from reasoning-style models), we parse the model's prediction from the content enclosed by the `<think>` tag and use it as the final answer for scoring.

### C.3. SAP Details

This appendix provides implementation details for the SAP intervention in Sec. 4.3, including (i) the three SAP modes and (ii) how we select affected layers and heads for each architecture.

**SAP modes.** Shared Attention Prior (SAP) replaces the original attention scores with a lightweight prior that is computed once per input and then shared across selected layers and heads, requiring substantially less computation than per-layer score estimation. We instantiate three SAP modes:

*(i) Visual-encoder attention.* Since the vision encoder is trained with vision-centric objectives (e.g., hierarchical vision encoders such as Swin Transformer (Liu et al., 2021)), we replace decoder attention scores with the last-layer attention maps from the visual encoder as a natural alignment prior. Note that the visual tokens used by the decoder may be merged relative to the encoder output (e.g., `spatial_merge_size`=2 in Qwen-style encoders (Team, 2025)), so we align resolutions by average pooling the encoder attention over each $m \times m$ merged block (with $m = \texttt{spatial\_merge\_size}$) before substitution.

*(ii) Patch complexity.* We compute a low-cost patch prior from the input image using the decoder patch size. For each patch $p$, we first convert RGB to grayscale (BT et al., 2011)

$$g(u,v) = 0.299\, R(u,v) + 0.587\, G(u,v) + 0.114\, B(u,v),$$

then define an efficient gradient-magnitude statistic via mean absolute finite differences:

$$G_x(p) = \frac{1}{HW'} \sum_{u=1}^{H} \sum_{v=1}^{W-1} \big|g(u,v+1) - g(u,v)\big|, \quad G_y(p) = \frac{1}{H'W} \sum_{u=1}^{H-1} \sum_{v=1}^{W} \big|g(u+1,v) - g(u,v)\big|,$$

$$\mathrm{grad}(p) = G_x(p) + G_y(p), \qquad \mathrm{var}(p) = \mathrm{Var}\big(g(u,v)\big), \qquad c(p) = \mathrm{grad}(p) + \mathrm{var}(p).$$

Here $H$ and $W$ denote the patch height and width (in pixels), and we set $H' = H - 1$ and $W' = W - 1$ to match the valid ranges of the finite differences. Intuitively, $\mathrm{grad}(p)$ summarizes local edge strength within the patch (Pertuz et al., 2013), while $\mathrm{var}(p)$ measures within-patch intensity dispersion; we combine them as $c(p)$ and use $\{c(p)\}$ as a patch-wise attention prior.

*(iii) Noise.* We directly sample a Gaussian tensor with the same shape as the target attention scores and substitute it as the prior.

**Selecting layers and heads.** We choose affected layers and heads via ablations (see Appendix D) for each architecture. Layers are selected by depth order (contiguous ranges), while heads are selected by ranking their *non-visual* attention mass. Concretely, let $A_{b,t,i}^{l,h}$ denote the normalized attention weight at layer $l$, head $h$, batch item $b$, for query position $t$ over key position $i$. Let the visual-token span be $[v_{\text{start}}, v_{\text{end}})$. We define the non-visual index set

$$\mathcal{N} = \{1, \ldots, v_{\text{start}} - 1\} \cup \{v_{\text{end}}, \ldots, S_c\}.$$

Using the last query position $t = -1$ (the current decoding step), we score each head by the negative mean non-visual mass:

$$s_h = -\frac{1}{B\,|\mathcal{N}|} \sum_{b=1}^{B} \sum_{i \in \mathcal{N}} A_{b,-1,i}^{l,h}.$$

We rank heads by $s_h$ and select a percentile band (e.g., $[0.0, 0.3]$) per chosen layer; for heads within this band, we replace their attention scores with the shared SAP prior. Our head-selection strategy is motivated by the empirically supported *head specialization* hypothesis in multimodal Transformers: different heads and layers tend to preferentially route modality-specific signals (e.g., visual vs. textual attributes) (Basile et al., 2025). To better decouple visual interactions from text-dominated routing effects, we rank heads by their *non-visual* attention mass and intervene on a chosen percentile range, so that the replacement primarily targets heads that allocate relatively less probability to non-visual tokens.

# D. Additional Results

In this section, we use VMC-Bench (Zhang et al., 2025c), which provides a comprehensive evaluation of LVLMs along five dimensions: General, Reasoning, OCR, Math, and Doc&Chart.

## D.1. Ablation Studies for SAP

We conduct ablations across all models; except that the mode ablation is already reported in Table 2, we focus here on ablating (i) the affected layers and (ii) the affected heads.

*Table 4.* Ablation Study on **Attention Heads** (Part I): Evaluation of General Perception and Reasoning Capabilities across Different Parameter Settings. The default configurations for each model is highlighted in bold red.

| Model | Heads | General | | | | | Reasoning | | | | | |
|---|---|---|---|---|---|---|---|---|---|---|---|---|
| | | VQAv2 | VizWiz | OKVQA | MMVet | A-OKVQA | MMStar | SEED | SciQA | RWQA | MMMU | GQA |
| Qwen-2.5-VL-7B | [0.0, 0.3] | 83.56 | 82.60 | 84.94 | 71.22 | 78.82 | 59.86 | 74.81 | 80.32 | 53.21 | 54.09 | 81.42 |
| | **[0.3, 0.6]** | **90.51** | **87.99** | **90.12** | 73.38 | **86.35** | **63.18** | 79.01 | 83.48 | **59.40** | **55.77** | **85.57** |
| | [0.6, 0.9] | 89.35 | 87.50 | 89.63 | 71.94 | 84.24 | 62.00 | 78.02 | **84.39** | 57.34 | 55.53 | 83.62 |
| | [0.2, 0.8] | 89.58 | 86.76 | 88.64 | 72.66 | 84.00 | 61.28 | **79.75** | 84.16 | 58.72 | 53.12 | 84.84 |
| | [0.0, 1.0] | 84.72 | 87.50 | 84.94 | **75.54** | 79.76 | 57.48 | 76.05 | 81.00 | 55.96 | 50.96 | 79.71 |
| LLaVA-1.5-7B | [0.0, 0.3] | 67.13 | 64.22 | 74.57 | 46.76 | 67.76 | 34.44 | 56.54 | 56.56 | **37.61** | 36.54 | 64.06 |
| | [0.3, 0.6] | **71.30** | 66.91 | 80.25 | 55.40 | 71.06 | 35.63 | 60.99 | **59.73** | 36.93 | **36.54** | 69.19 |
| | [0.6, 0.9] | 68.52 | 67.16 | 79.01 | 49.64 | 68.24 | 32.78 | 59.01 | 58.60 | 36.70 | 34.62 | 67.24 |
| | **[0.2, 0.8]** | **71.30** | **72.06** | **81.73** | 53.24 | **73.41** | **38.24** | **63.46** | 56.79 | 36.24 | 35.10 | **70.17** |
| | [0.0, 1.0] | 66.44 | 67.89 | 76.05 | 46.76 | 66.59 | 31.59 | 53.58 | 50.68 | 36.47 | 33.17 | 60.88 |
| LLaVA-OV-7B | [0.0, 0.3] | 51.16 | 57.11 | 62.96 | 47.48 | 58.35 | 41.57 | 51.36 | 51.81 | 42.66 | 35.34 | 55.75 |
| | **[0.3, 0.6]** | 83.33 | **85.29** | **88.15** | **68.35** | 85.88 | **52.97** | 77.04 | **81.67** | 54.13 | **43.27** | **84.60** |
| | [0.6, 0.9] | 83.80 | 84.56 | 87.16 | 67.63 | **86.82** | 52.02 | 76.54 | 81.00 | 55.96 | 42.31 | **84.60** |
| | [0.2, 0.8] | **84.72** | 83.82 | 84.69 | 64.75 | **86.82** | 49.17 | **79.26** | 77.60 | **56.42** | 41.83 | 82.89 |
| | [0.0, 1.0] | 31.71 | 31.37 | 31.11 | 23.02 | 31.06 | 28.98 | 29.38 | 32.81 | 29.36 | 29.09 | 31.05 |

The head ablation results are reported in Tables 4 and 5. Overall, intervening on mid-quantile heads consistently outperforms modifying either tail, while the models are more sensitive to perturbations on the lower-quantile heads. Under our head partition criterion, these lower-quantile heads primarily attend to non-visual (text) tokens; altering them therefore disrupts textual representations and degrades performance. For reference, we highlight the default affected-head setting for each architecture in red in the tables.

The layer ablation results are reported in Tables 6 and 7. We observe that LVLMs are highly sensitive to interventions in early layers, whereas perturbing middle or late layers typically causes only minor changes. For instance, for LLaVA-1.5-7B,

*Table 5.* Ablation Study on **Attention Heads** (Part II): Performance Analysis on Document/Chart and OCR Task. The default configurations are highlighted in bold red. AVG represents the overall average across all benchmarks, including Table 4.

| Model | Heads | Math | | Doc & Chart | | | | | OCR | | AVG |
|---|---|---|---|---|---|---|---|---|---|---|---|
| | | Vista | Vision | DocVQA | Table | ChartQA | InfoVQA | AI2D | TextVQA | OCRVQA | |
| Qwen-2.5-VL-7B | [0.0, 0.3] | 51.49 | 32.36 | 72.61 | 66.22 | 74.77 | 55.53 | 72.89 | 93.03 | 83.16 | 70.35 |
| | **[0.3, 0.6]** | 53.96 | **34.38** | **77.06** | **72.07** | **79.59** | **59.22** | 77.22 | 95.28 | 91.71 | **74.76** |
| | [0.6, 0.9] | 53.47 | 33.48 | 75.95 | 70.72 | 78.67 | 57.83 | **77.68** | **95.73** | **94.04** | 74.06 |
| | [0.2, 0.8] | 53.96 | 33.26 | 72.61 | 70.27 | 79.36 | 58.76 | 77.45 | 93.93 | **94.04** | 73.86 |
| | [0.0, 1.0] | **55.45** | 32.13 | 69.27 | 62.39 | 74.77 | 53.92 | 73.58 | 91.69 | 92.75 | 70.98 |
| LLaVA-1.5-7B | [0.0, 0.3] | 22.77 | 25.39 | 34.30 | 25.68 | 26.61 | 30.41 | 43.28 | 55.96 | 65.03 | 46.78 |
| | [0.3, 0.6] | 25.74 | 28.54 | 37.19 | 29.05 | 32.34 | 29.95 | 41.91 | 61.35 | 67.36 | 49.87 |
| | [0.6, 0.9] | **29.70** | 26.97 | 38.31 | 27.93 | 30.28 | 29.26 | **43.96** | 61.12 | 68.91 | 48.90 |
| | **[0.2, 0.8]** | 25.25 | 26.97 | **38.75** | **29.50** | **32.80** | 31.80 | 42.82 | **63.37** | **70.98** | **50.70** |
| | [0.0, 1.0] | 26.73 | **30.34** | 33.85 | 27.70 | 28.67 | **32.95** | 41.91 | 60.67 | 68.13 | 47.05 |
| LLaVA-OV-7B | [0.0, 0.3] | 45.05 | 29.66 | 47.88 | 35.81 | 40.37 | 33.41 | 43.51 | 52.36 | 61.66 | 47.26 |
| | **[0.3, 0.6]** | 51.49 | **30.79** | 71.49 | 47.75 | **60.09** | 46.54 | 65.83 | **87.42** | 89.90 | **67.80** |
| | [0.6, 0.9] | **53.47** | 29.66 | 72.83 | **48.42** | 56.65 | **47.47** | **66.29** | **87.42** | 90.67 | 67.76 |
| | [0.2, 0.8] | 49.01 | 28.99 | **73.05** | 45.50 | 54.13 | 46.77 | 63.55 | 85.17 | 87.56 | 66.29 |
| | [0.0, 1.0] | 31.68 | 22.25 | 33.63 | 26.80 | 32.57 | 31.11 | 27.56 | 31.69 | 36.53 | 30.14 |

*Table 6.* Ablation Study on **Affected Layers** (Part I): Evaluation of General Perception and Reasoning Capabilities across Different Layer Configurations. The default settings are highlighted in bold red.

| Model | Layers | General | | | | | Reasoning | | | | | |
|---|---|---|---|---|---|---|---|---|---|---|---|---|
| | | VQAv2 | VizWiz | OKVQA | MMVet | A-OKVQA | MMStar | SEED | SciQA | RWQA | MMMU | GQA |
| LLaVA-1.5-7B | [2, 7] | 46.30 | 42.40 | 49.88 | 41.73 | 47.53 | 31.83 | 42.22 | 38.69 | 28.21 | 28.12 | 46.45 |
| | [2, 13] | 36.81 | 34.31 | 39.26 | 28.78 | 32.47 | 26.37 | 30.86 | 36.20 | 30.73 | 23.80 | 35.21 |
| | [6, 11] | 58.33 | 51.96 | 61.23 | 44.60 | 56.24 | 28.27 | 43.21 | 43.89 | 27.29 | 32.93 | 47.19 |
| | [12, 17] | 68.52 | 68.87 | 73.58 | 46.04 | 67.06 | 32.54 | 57.04 | 52.71 | 35.09 | 35.34 | 63.57 |
| | [14, 25] | 71.30 | 66.42 | 77.78 | 49.64 | 70.35 | 34.68 | 60.99 | 58.37 | 38.30 | 36.06 | **71.88** |
| | **[18, 23]** | 71.30 | **72.06** | **81.73** | **53.24** | **73.41** | **38.24** | **63.46** | 56.79 | 36.24 | 35.10 | 70.17 |
| | [18, 29] | **72.92** | 65.93 | 79.51 | 47.48 | 70.35 | 34.44 | 58.77 | 57.69 | 36.24 | 34.62 | 70.42 |
| | [22, 31] | 67.82 | 68.14 | 77.53 | 47.48 | 67.76 | 31.35 | 59.51 | 57.24 | 36.70 | **36.54** | 66.99 |
| | [24, 29] | 69.91 | 66.18 | 77.28 | 45.32 | 70.82 | 33.02 | 63.21 | **58.60** | **39.68** | 34.62 | 68.22 |
| LLaVA-OV-7B | [0, 6] | 65.97 | 68.14 | 70.12 | 44.60 | 65.41 | 40.86 | 56.05 | 66.06 | 43.12 | 35.58 | 71.64 |
| | [0, 13] | 59.95 | 60.78 | 62.72 | 40.29 | 58.82 | 40.38 | 54.32 | 59.73 | 33.49 | 30.77 | 60.64 |
| | [7, 13] | 82.87 | 81.62 | 83.95 | 61.87 | 86.35 | 47.98 | **77.28** | 78.28 | 54.13 | 40.14 | 80.44 |
| | [14, 20] | **84.03** | **87.01** | 86.91 | 63.31 | 86.59 | 48.93 | 75.80 | 80.09 | 52.29 | 42.07 | 82.15 |
| | [14, 27] | 83.33 | 83.09 | 86.42 | 64.03 | **87.06** | **52.97** | 76.30 | 81.00 | **54.59** | **43.99** | **85.82** |
| | **[21, 27]** | 83.33 | 85.29 | **88.15** | **68.35** | 85.88 | **52.97** | 77.04 | **81.67** | 54.13 | 43.27 | 84.60 |

*Table 7.* Ablation Study on **Affected Layers** (Part II): Performance Analysis on Document/Chart Understanding and OCR Task. The default settings are highlighted in bold red.

| Model | Layers | Math | | Doc & Chart | | | | | OCR | | |
|---|---|---|---|---|---|---|---|---|---|---|---|
| | | Vista | Vision | DocVQA | Table | ChartQA | InfoVQA | AI2D | TextVQA | OCRVQA | AVG |
| LLaVA-1.5-7B | [2, 7] | 25.25 | 24.27 | 30.51 | 27.25 | 26.83 | 29.72 | 32.57 | 44.04 | 43.01 | 36.34 |
| | [2, 13] | 28.71 | 28.76 | 25.17 | 23.42 | 27.75 | 23.27 | 28.02 | 38.88 | 31.61 | 30.52 |
| | [6, 11] | 27.23 | 26.52 | 35.63 | 24.77 | 24.31 | **34.10** | 37.13 | 48.31 | 52.85 | 40.30 |
| | [12, 17] | 23.76 | 27.42 | 34.52 | 29.50 | 26.15 | 31.80 | 38.95 | 60.45 | 65.54 | 46.92 |
| | [14, 25] | 25.74 | 27.42 | 37.42 | **31.98** | 28.67 | 31.34 | 42.82 | 61.80 | 66.58 | 49.48 |
| | **[18, 23]** | 25.25 | 26.97 | 38.75 | 29.50 | 32.80 | 31.80 | 42.82 | **63.37** | **70.98** | **50.70** |
| | [18, 29] | 28.71 | 28.54 | **41.43** | 29.50 | 26.15 | 28.80 | 42.14 | 62.47 | 65.80 | 49.10 |
| | [22, 31] | **35.15** | **30.11** | 40.09 | 31.31 | 29.59 | 31.11 | 42.60 | 60.90 | 68.65 | 49.33 |
| | [24, 29] | 28.22 | 24.27 | 37.19 | 27.03 | **33.26** | 30.65 | **45.10** | 59.10 | 67.36 | 48.95 |
| LLaVA-OV-7B | [0, 6] | 37.13 | 27.42 | 49.89 | 38.06 | 44.95 | 31.80 | 51.71 | 64.27 | 68.13 | 52.05 |
| | [0, 13] | 37.62 | 28.09 | 44.99 | 32.21 | 35.55 | 36.87 | 46.01 | 55.51 | 68.39 | 47.36 |
| | [7, 13] | 53.96 | 30.11 | 71.94 | 42.57 | 50.46 | **47.24** | 64.24 | 84.72 | 87.31 | 65.37 |
| | [14, 20] | **54.46** | 28.31 | **73.50** | 46.62 | 52.52 | 43.32 | 64.69 | 84.49 | 88.86 | 66.30 |
| | [14, 27] | 49.50 | 27.42 | 71.94 | 45.72 | 58.94 | 45.16 | 65.15 | **87.42** | 89.12 | 66.95 |
| | **[21, 27]** | 51.49 | **30.79** | 71.49 | **47.75** | **60.09** | 46.54 | **65.83** | **87.42** | **89.90** | **67.80** |

intervening on Layers 1–7 reduces accuracy by 13%, while intervening on Layers 22–31 incurs only a 0.5% drop. This pattern further supports a pervasive issue in current LVLMs: a substantial fraction of decoder attention computation is redundant.

### D.2. Extending SAP to Other Architectures and Larger Variants

Table 8 shows that head-percentile interventions yield consistent, model-dependent optima across Qwen-2.5-VL variants under the same affected-layer range ($[1, 27]$). In particular, mid-percentile heads (e.g., $[0.3, 0.6]$) are frequently the best-performing choice for several variants, while extreme ranges can be substantially less stable for some models. Overall, these results indicate that the sensitivity of SAP-style interventions is structured rather than uniform across heads, motivating architecture-aware head selection in subsequent experiments.

*Table 8.* Head ablation on Qwen-2.5-VL architecture variants (**affected layers fixed to** $[1, 27]$ on the decoder). Each column reports VMC accuracy under a head percentile interval $[h_{\min}, h_{\max}]$; the best setting per model is **bolded**.

| Model | [0.0, 0.3] | [0.3, 0.6] | [0.6, 0.9] | [0.2, 0.8] | [0.0, 1.0] |
|---|---|---|---|---|---|
| CoF-rl-model-7b | 0.567 | **0.637** | 0.631 | 0.613 | 0.461 |
| CoF-sft-model-7b | 0.558 | **0.640** | 0.634 | 0.595 | 0.415 |
| MM-Eureka-Qwen-32B | **0.727** | 0.706 | 0.676 | 0.457 | 0.161 |
| MM-Eureka-Qwen-7B | 0.531 | 0.623 | **0.627** | 0.436 | 0.151 |
| Ocean_R1_7B_Instruct | **0.639** | 0.594 | 0.499 | 0.248 | 0.086 |
| Orsta-7B | 0.537 | **0.604** | 0.599 | 0.309 | 0.087 |
| Qwen2.5-VL-32B-Instruct | 0.827 | **0.829** | 0.828 | 0.816 | 0.791 |

## E. Layer-wise Attention Tracing

**Tracing cross-patch interactions.** We provide a visualization tool to trace layer-wise visual interactions from decoder attention. For each layer $l$, we construct a visual interaction graph $\mathcal{G}^{(l)} = (\mathcal{V}, \mathcal{E}^{(l)})$ over visual patches (Abnar & Zuidema, 2020), where $\mathcal{V} = \{1, \ldots, S_v\}$ indexes visual tokens and edges are induced by thresholded visual-to-visual attention. Let $\mathbf{A}^{(l)} \in [0, 1]^{S \times S}$ denote the head-averaged attention matrix at layer $l$ (after averaging over heads). Restricting to the visual block yields $\mathbf{A}_{vv}^{(l)} \in [0, 1]^{S_v \times S_v}$. We include a directed edge $j \to i$ whenever

$$A_{vv}^{(l)}(i, j) \geq \tau, \qquad \tau = 0.1,$$

interpreting $A_{vv}^{(l)}(i, j)$ as patch $i$ attending to patch $j$.

**Constructing key regions from COCO instance annotations.** To operationalize question-relevant visual evidence, we leverage the fact that POPE samples are drawn from MSCOCO images and thus inherit COCO instance-level object annotations with localization information (e.g., bounding boxes) (Lin et al., 2014). For each POPE query, we identify the referenced object category and retrieve its annotated bounding box(es). After applying the same image preprocessing as the LVLM (e.g., resizing and patchification into a $t_h \times t_w$ visual grid), we map each bounding box to a set of visual patch indices by marking all patches whose spatial support intersects the box. The union of these patches forms the key-patch set $\mathcal{K} \subseteq \mathcal{V} = \{1, \dots, S_v\}$, which we use below to quantify how much of the layer-wise visual interaction graph is routed through question-relevant regions.

**Key-region degree ratio.** For each layer, we treat $\mathcal{G}^{(l)}$ as the visual interaction graph and quantify how much interaction mass is routed through question-relevant regions. Let $\mathcal{K} \subseteq \mathcal{V}$ be the set of key patches that correspond to question-relevant visual evidence. Define the key-region degree ratio as

$$\rho^{(l)} = \frac{\left|\{(j \to i) \in \mathcal{E}^{(l)} : i \in \mathcal{K} \text{ or } j \in \mathcal{K}\}\right|}{|\mathcal{E}^{(l)}|}.$$

We randomly sampled 100 correctly answered cases and 100 incorrectly answered cases, and computed $\rho^{(l)}$ for each case; see Figures 5–10 for case studies. Averaged across samples, the key-region degree ratio is $4.2\%$ for incorrect answers versus $13.1\%$ for correct answers, indicating that failures are associated with substantially weaker attention-mediated interaction around question-relevant visual evidence, consistent with systematic misallocation of visual attention.

## F. Theorem and Proofs

**Lemma F.1** (Range of $\Delta\mathcal{S}$, $\Delta\mathcal{D}$, and RID). *For $\mathbf{X}, \mathbf{X}' \in \mathbb{R}^{S \times H}$, we have $\Delta\mathcal{S}(\mathbf{X} \mid \mathbf{X}') \in [0, 1]$ and $\Delta\mathcal{D}(\mathbf{X} \mid \mathbf{X}') \in [0, 1]$. Consequently, $\mathrm{RID}(\mathbf{X} \mid \mathbf{X}') \in [0, 2]$.*

*Proof.* Since $\mathrm{eRank}(\mathbf{Z}) \in [1, \min\{S, H\}]$ for any $\mathbf{Z}$, we have $0 \leq |\mathrm{eRank}(\mathbf{X}') - \mathrm{eRank}(\mathbf{X})| \leq \min\{S, H\}$, hence

$$\Delta\mathcal{S}(\mathbf{X} \mid \mathbf{X}') = \frac{\left|\mathrm{eRank}(\mathbf{X}') - \mathrm{eRank}(\mathbf{X})\right|}{\min\{S, H\}} \in [0, 1].$$

Let $\mathbf{P}$ be any orthogonal projector. Then $\mathbf{I} - \mathbf{P}$ is also an orthogonal projector and is non-expansive: $\|(\mathbf{I} - \mathbf{P})\mathbf{Z}\|_F \leq \|\mathbf{Z}\|_F$ and $\|\mathbf{Z}(\mathbf{I} - \mathbf{P})\|_F \leq \|\mathbf{Z}\|_F$. Applying this with $\mathbf{P} = \mathbf{P}_{\mathcal{C}(\mathbf{X})}$ and $\mathbf{P} = \mathbf{P}_{\mathcal{R}(\mathbf{X})}$ yields

$$\left\|(\mathbf{I} - \mathbf{P}_{\mathcal{C}(\mathbf{X})})\mathbf{X}'\right\|_F + \left\|\mathbf{X}'(\mathbf{I} - \mathbf{P}_{\mathcal{R}(\mathbf{X})})\right\|_F \leq 2\|\mathbf{X}'\|_F.$$

Therefore, under the normalization by $2\|\mathbf{X}'\|_F$,

$$\Delta\mathcal{D}(\mathbf{X} \mid \mathbf{X}') = \frac{\left\|(\mathbf{I} - \mathbf{P}_{\mathcal{C}(\mathbf{X})})\mathbf{X}'\right\|_F + \left\|\mathbf{X}'(\mathbf{I} - \mathbf{P}_{\mathcal{R}(\mathbf{X})})\right\|_F}{2\|\mathbf{X}'\|_F} \in [0, 1].$$

Finally, $\mathrm{RID} = \Delta\mathcal{S} + \Delta\mathcal{D}$ gives $\mathrm{RID}(\mathbf{X} \mid \mathbf{X}') \in [0, 2]$. $\qquad\square$

**Theorem F.2** (Eckart–Young–Mirsky Theorem (Eckart & Young, 1936)). *Let $\mathbf{X}$ have SVD as in Definition 3.3. For any $k \leq Q$, define the rank-$k$ truncation*

$$\mathbf{X}_k = \sum_{i=1}^{k} \sigma_i \mathbf{u}_i \mathbf{v}_i^\top.$$

*Then $\mathbf{X}_k$ solves the best rank-$k$ approximation problem under the Frobenius norm:*

$$\mathbf{X}_k \in \arg\min_{\mathrm{rank}(\mathbf{Y}) \leq k} \|\mathbf{X} - \mathbf{Y}\|_F.$$

**Theorem F.3** (Expectation Equivalence under Attention Noise Injection). *Scenario 1: random* **QKV**. *Consider an attention head with $N$ key-value pairs. Let $Q_{noise}, K_{noise}, V_{noise}$ be the random Gaussian replacements for the original query, key, value matrices, where each has the same mean and variance as the original $Q, K, V$ respectively. The attention output in scenario (1) (replacing $Q, K, V$ by noise) for a single query can be written as a weighted sum of the value vectors:*

$$Y_{noise} = \sum_{i=1}^{N} a_i \, v_i^{(noise)},$$

*where $v_i^{(noise)}$ is the $i$-th row of $V_{noise}$ and $a_i$ is the attention weight for key $i$ given by the softmax:*

$$a_i = \frac{\exp\big((q^{(noise)})^\top k_i^{(noise)}/\sqrt{d}\big)}{\sum_{j=1}^{N} \exp\big((q^{(noise)})^\top k_j^{(noise)}/\sqrt{d}\big)},$$

*with $q^{(noise)}$ the query vector and $k_i^{(noise)}$ the $i$-th key (row of $K_{noise}$). By construction of the softmax, $\sum_{i=1}^{N} a_i = 1$ for any realization. Under the assumption that the random keys $(k_1^{(noise)}, \ldots, k_N^{(noise)})$ are i.i.d. (making all key positions statistically symmetric), the attention weights $\{a_i\}$ are an exchangeable set. In particular, by symmetry we have $\mathbb{E}[a_i] = \frac{1}{N}$ for each $i$. Now taking expectation of $Y_{noise}$ (over the random $Q_{noise}, K_{noise}, V_{noise}$) and using the law of total expectation, we get:*

$$\mathbb{E}[Y_{noise}] = \mathbb{E}\Big[\sum_{i=1}^{N} a_i \, v_i^{(noise)}\Big] = \mathbb{E}\Big[\mathbb{E}\big[\sum_{i=1}^{N} a_i \, v_i^{(noise)} \mid V_{noise}\big]\Big].$$

*Conditioning on the random values $V_{noise} = \{v_i^{(noise)}\}_{i=1}^{N}$, the attention weights are independent of $V_{noise}$ and still satisfy $\mathbb{E}[a_i \mid V_{noise}] = \frac{1}{N}$. Thus*

$$\mathbb{E}\Big[\sum_{i=1}^{N} a_i \, v_i^{(noise)} \,\Big|\, V_{noise}\Big] = \sum_{i=1}^{N} \mathbb{E}[a_i \mid V_{noise}] v_i^{(noise)} = \frac{1}{N} \sum_{i=1}^{N} v_i^{(noise)}.$$

*The right-hand side is simply the average of the $N$ i.i.d. random value vectors. Therefore, its expectation is the mean of the $V_{noise}$ distribution:*

$$\mathbb{E}[Y_{noise}] = \mathbb{E}\Big[\frac{1}{N} \sum_{i=1}^{N} v_i^{(noise)}\Big] = \mathbb{E}[v_i^{(noise)}] = \mu_V,$$

*where $\mu_V$ denotes the mean of the original $V$ (and $V_{noise}$) distribution.*

*Scenario 2: random* $\Delta$. *In scenario (2), where we directly replace the final attention output with Gaussian noise of the same distribution as the true $Y = AV$ (with $A$ the attention matrix), the injected output $Y_{direct}$ is a Gaussian random vector with mean set to $\mu_Y$, the mean of the original attention output. Typically, if the original model's parameters are approximately zero-mean (as is common in weight initialization), the distribution of the true attention output $Y$ will have mean $\mu_Y \approx 0$. In our case above, we found $\mu_Y = \mu_V$, since the attention mechanism produces a convex combination of the values. Under the assumption that the original attention output's mean $\mu_Y$ equals $\mu_V$ (which holds, for example, if weights are zero-centered so that queries and keys induce no bias in attention, or more generally under the symmetry argument given), we have*

$$\mathbb{E}[Y_{direct}] = \mu_Y = \mu_V = \mathbb{E}[Y_{noise}].$$

*Thus, the mean of the noise-injected output in scenario (1) is the same as the mean of the direct noise output in scenario (2). In other words, both replacement strategies produce outputs with the same expected mean.*

**Theorem F.4** (Manifold Coincidence Theorem for RID). *We aim to show that if $\mathrm{RID}(X \mid X') = 0$, then $X$ and $X'$ share the same manifold structure – in particular, $X'$ lies in the same underlying subspace as $X$ with equivalent spectral complexity. By the definition of **Representation Information Discrepancy (RID)**, we have*

$$\mathrm{RID}(X \mid X') = \Delta\mathcal{S}(X \mid X') + \Delta\mathcal{D}(X \mid X').$$

*The condition $\mathrm{RID}(X \mid X') = 0$ necessitates that both non-negative components vanish: $\Delta\mathcal{S}(X \mid X') = 0$ and $\Delta\mathcal{D}(X \mid X') = 0$.*

*Firstly, the condition $\Delta\mathcal{S}(X \mid X') = 0$ implies the invariance of the spectral complexity as measured by the effective rank. Since the effective rank serves as a continuous proxy for the number of active degrees of freedom, its conservation indicates that the intrinsic dimensionality of the representation remains unchanged. Under the manifold hypothesis characterizing* $\mathbf{X}$*, this implies that the algebraic rank is preserved, i.e.,* $\mathrm{rank}(X') = \mathrm{rank}(X) = r$*. Consequently, both matrices reside within the same fixed-rank manifold geometry* $\mathcal{M}_r$*.*

*Secondly,* $\Delta\mathcal{D}(X \mid X') = 0$ *signifies that* $X'$ *introduces no new* information support *relative to* $X$*. By the definition of support innovation, the projection residuals must be zero:*

$$\left\|(I - \mathbf{P}_{\mathcal{C}(X)})\,X'\right\|_F = 0, \qquad \left\|X'\,(I - \mathbf{P}_{\mathcal{R}(X)})\right\|_F = 0,$$

*where* $\mathbf{P}_{\mathcal{C}(X)}$ *and* $\mathbf{P}_{\mathcal{R}(X)}$ *are the orthogonal projectors onto the column space* $\mathcal{C}(X)$ *and row space* $\mathcal{R}(X)$ *of* $X$*, respectively. These conditions are algebraically equivalent to:*

$$\mathcal{C}(X') \subseteq \mathcal{C}(X), \qquad \mathcal{R}(X') \subseteq \mathcal{R}(X).$$

*Having established that* $\mathrm{rank}(X') = \mathrm{rank}(X) = r$*, it follows that* $\dim(\mathcal{C}(X')) = \dim(\mathcal{C}(X)) = r$*. A fundamental result in linear algebra states that if a subspace* $\mathcal{V}$ *is contained in a subspace* $\mathcal{W}$ *of the same finite dimension, then* $\mathcal{V} = \mathcal{W}$*. Therefore, we conclude:*

$$\mathcal{C}(X') = \mathcal{C}(X), \qquad \mathcal{R}(X') = \mathcal{R}(X).$$

*This proves that* $X'$ *shares exactly the same left and right singular vector subspaces as* $X$*, meaning the* information support *is identical:* $\mathcal{D}_{X'} = \mathcal{D}_X$*. Combined with the unchanged spectrum (*$\mathcal{S}_{X'} = \mathcal{S}_X$*), we have*

$$\mathcal{I}(X') \;=\; (\mathcal{S}_{X'}, \mathcal{D}_{X'}) \;=\; (\mathcal{S}_X, \mathcal{D}_X) \;=\; \mathcal{I}(X).$$

*In conclusion, when* $\mathrm{RID}(X \mid X') = 0$*,* $X'$ *contains no new representation information compared to* $X$*. Geometrically,* $X$ *and* $X'$ *coincide in the manifold parameterization: they possess the same rank and occupy the same supporting subspaces. Thus,* $X$ *and* $X'$ ***share one manifold space***, *differing only by an internal reconfiguration of information within that shared subspace.*

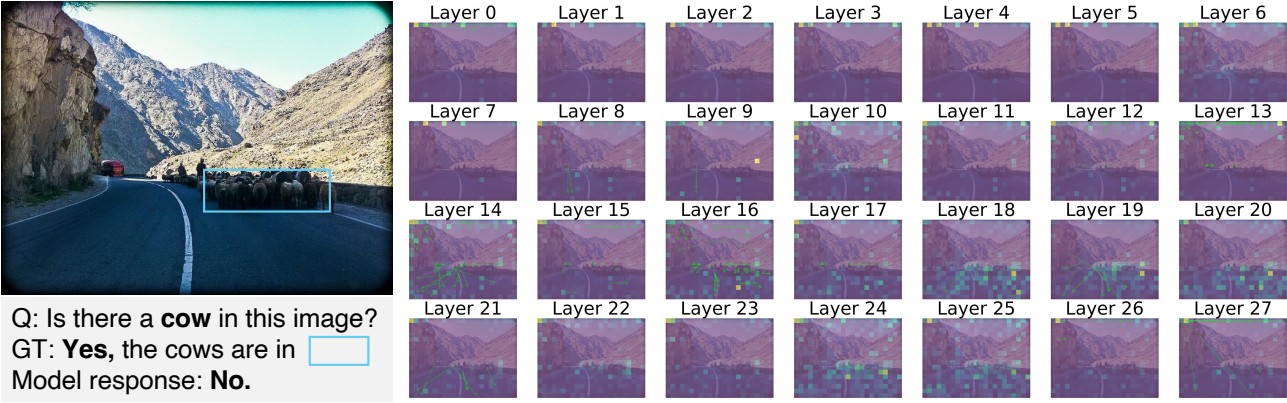

*Figure 5.* **Case 1.** Layer-wise visual attention tracing. Only layer 16 exhibits cross-patch interactions within the key region (cows).

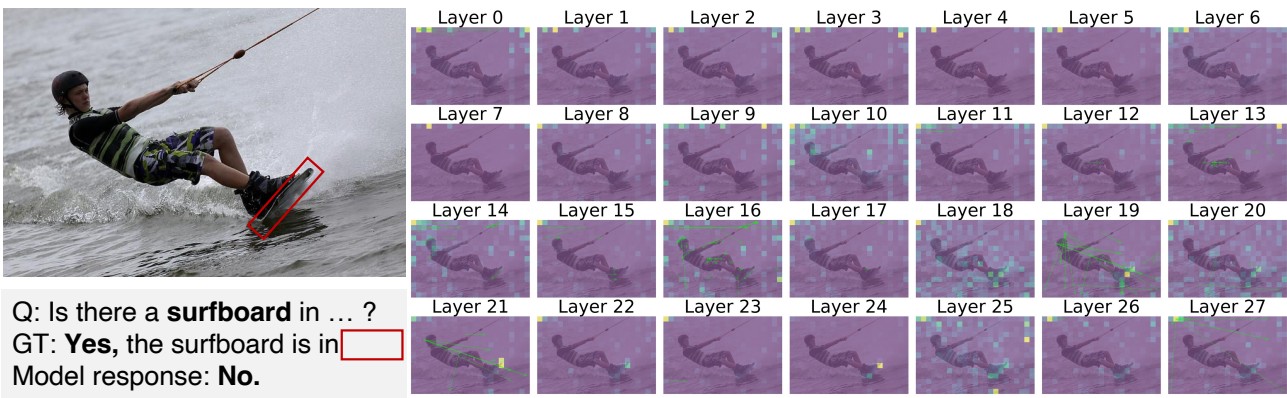

*Figure 6.* **Case 2.** Layer-wise visual attention tracing. Layer 16 and 19 exhibit cross-patch interactions within the key region (surfboard).

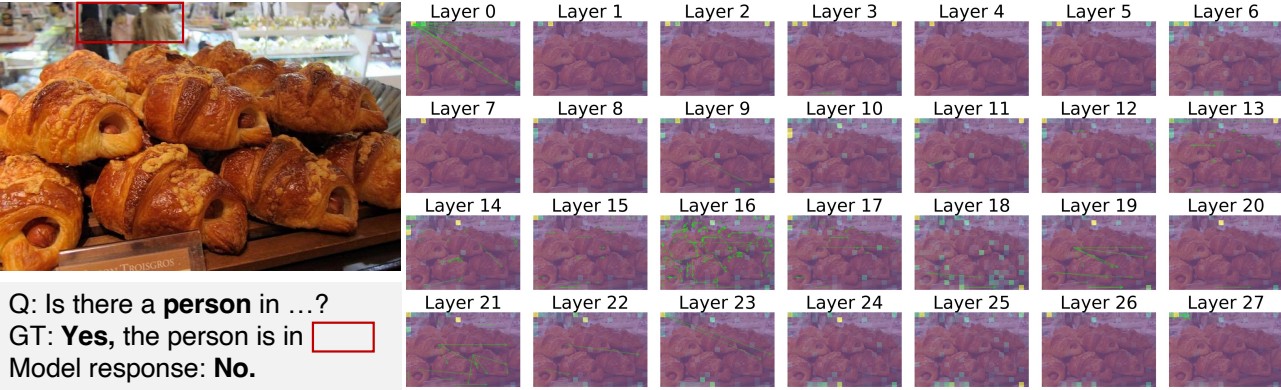

*Figure 7.* **Case 3.** Layer-wise visual attention tracing. Layer 16 and 17 exhibit cross-patch interactions within the key region (person).

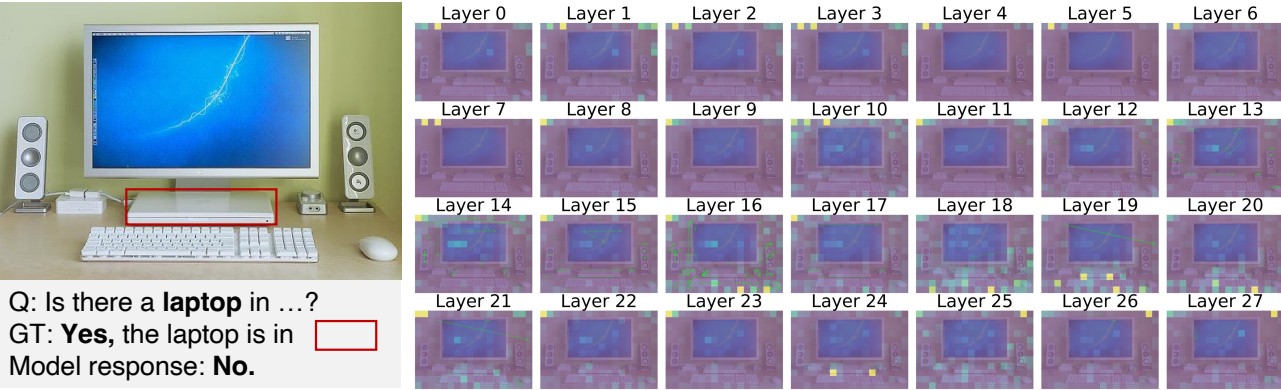

*Figure 8.* **Case 4.** Layer-wise visual attention tracing. Layer 13-17 exhibit cross-patch interactions within the key region (laptop).

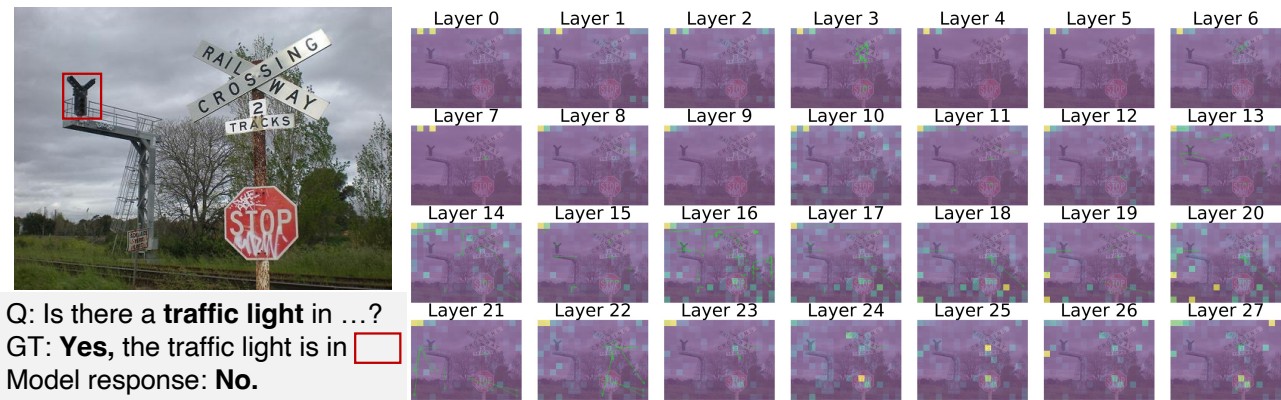

*Figure 9.* **Case 5.** Layer-wise visual attention tracing. Layer 16 and 17 exhibit cross-patch interactions within the key region (traffic light).

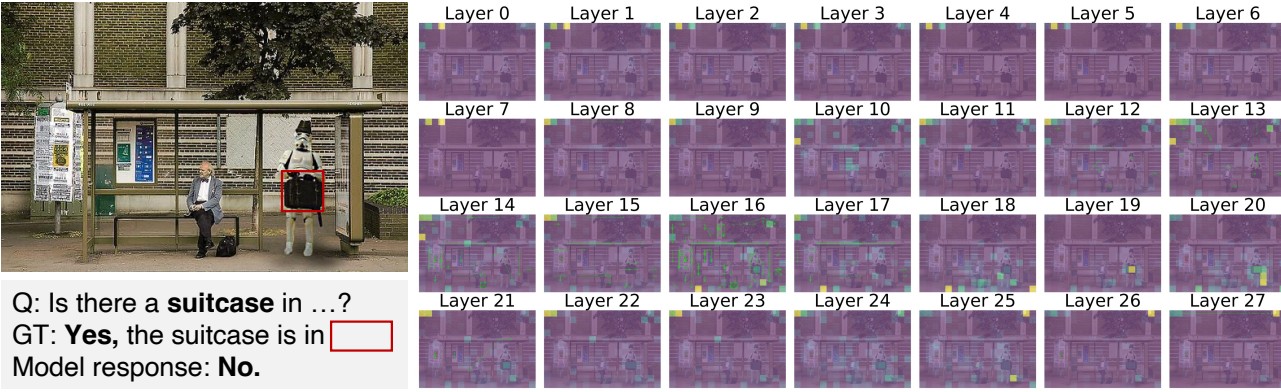

*Figure 10.* **Case 6.** Layer-wise visual attention tracing. Layer 13, 15 and 16 exhibits cross-patch interactions within the key region (suitcase).

