# OpenReview forum: "Large Vision-Language Models Get Lost in Attention"
_ICML.cc/2026/Conference — ICML 2026 regular_

### Official Review · Reviewer_qfjM · 2026-03-02

**Soundness:** 3
**Presentation:** 3
**Significance:** 2
**Originality:** 3
**Overall Recommendation:** 4
**Confidence:** 2

**Summary:**

This paper proposes an information-theoretic and geometric framework for analyzing the functional roles of attention and FFN modules in LVLM decoders. It introduces two metrics—innovations (RID) and reconfiguration (MixIG)—to characterize how each module transforms representations. Empirically, the authors find that attention layers are predominantly reconfiguration-heavy, while FFNs are more innovation-heavy. Moreover, MixIG varies substantially across attention layers, suggesting that attention may be redundant in some layers; this hypothesis is supported by self-attention score replacement experiments.

**Compliance With Llm Reviewing Policy:**

Affirmed.

**Ethical Review Concerns:**

Thanks for the detailed rebuttal response from the authors. The metric to differentiate FFN and attention is interesting and meaningful. The authors also provide some results to prove their metric is unified for both language and vision modalities. Since I have given a positive score for this paper, I maintain my score.

**Final Justification:**

Thanks for the detailed rebuttal response from the authors. The metric to differentiate FFN and attention is interesting and meaningful. The authors also provide some results to prove their metric is unified for both language and vision modalities.
Due to I have given a positive score for this paper, I maintain my score.

**Key Questions For Authors:**

Please see weakness

**Limitations:**

The conclusion that attention can be redundant is interesting, but it is not entirely novel (e.g., MetaFormer, 2022). Moreover, self-attention remains widely used in vision transformers and LLMs. As a result, a simple experiment demonstrating attention redundancy may not be fully convincing on its own. A deeper analysis—clarifying which tasks attention is redundant for and which tasks truly require attention—would likely be more informative and insightful.

**Strengths And Weaknesses:**

# Strengths

1. The paper is interesting and timely, especially in its attempt to disentangle the distinct functions of attention and FFN modules in LVLM decoders using principled metrics.
2. The self-attention score replacement experiments are compelling: they provide a simple and direct validation of the claimed redundancy of attention in certain layers. This result is reminiscent of MetaFormer [1], which argues that attention is not strictly necessary in vision transformers; in contrast, this paper goes further by offering a more fine-grained functional analysis.
3. The paper is clearly written and well organized, and the main idea is communicated effectively.

[1] MetaFormer Is Actually What You Need for Vision

# Weaknesses

1. A major source of confusion is whether the analysis is conducted over *all* tokens or only vision tokens. In the method section, the notation \(X\) appears to refer to the full token sequence (including both vision and language tokens). However, in the self-attention score replacement experiments, only attention scores among vision tokens are replaced. This mismatch makes it unclear what exactly the proposed metrics measure, and how the experimental intervention aligns with the theoretical framing.
2. Regardless of whether the focus is on vision tokens alone or all tokens, I think the interaction between vision and language tokens is important in LVLMs—especially given the trend toward unified multimodal processing. It would strengthen the paper to analyze cross-modal interactions explicitly (e.g., vision-to-text and text-to-vision attention patterns) and to discuss whether the proposed conclusions and metrics generalize to text and other modalities.

---

> ### Author Rebuttal · Authors · 2026-03-30
>
> We sincerely thank you for your careful review and constructive suggestions. Below we provide point-by-point responses to your concerns and summarize the additional analyses and revisions we made to further clarify the token scope, theory–intervention alignment, and generalization to cross-modal and language settings.
>
> ---
> > `W1`: _Token scope and theory–intervention alignment_
>
> We apologize for the confusion caused by our presentation. Your understanding is correct. In the method section, $X$ denotes the **full token sequence**, including both vision and language tokens. Accordingly, the proposed metrics, RID and MixIG, are defined over all tokens and are used to analyze and disentangle the functional roles of attention and FFN at the module level, as shown in Fig. 2 and Obs. 1 to 2.
>
> After this full-token analysis, our logic is as follows. The layer-wise MixIG analysis (Fig. 3) suggests that, although attention is responsible for information interaction, token mixing in the middle attention blocks is often **weak**. This phenomenon is particularly evident in **visual** attention. We therefore further investigate the redundancy of visual attention. From a representation perspective, this weak mixing pattern provides **theoretical evidence** that a substantial portion of visual attention may be redundant.
>
> A more direct question, however, is whether this theoretically suggested redundancy in visual attention actually affects the final model output. This is precisely the motivation for SAP: to move from representation-level analysis to output-level validation through intervention. Surprisingly, we find that replacing a large portion of visual attention scores with noise does not harm the output, and can even improve it in some cases. This provides the **direct evidence** behind our redundancy claim.
>
> To further strengthen this connection, we additionally conduct a MixIG-guided SAP experiment. Specifically, we use MixIG to identify 50\% of the layers with the smallest interaction contribution and then apply SAP to these layers.
>
> |Model|General|Reasoning|OCR|Doc\&Chart|Overall|
> |-|-:|-:|-:|-:|-:|
> |Qwen-2.5-VL-3B|78.3|60.2|92.2|70.2|72.3|
> |+ fixed-SAP|78.7|60.9|92.9|70.2|72.6|
> |+ MixIG-SAP|80.2|61.8|93.5|71.7|73.9|
>
> These results show that MixIG-guided SAP is more effective than fixed-SAP, providing a clearer alignment between the theoretical analysis and the intervention results, and supporting our claim that the weak interactions identified by MixIG correspond to attention components that are largely replaceable in practice.
>
> ---
> > `W2`: _Generalization to cross-modal interaction and other modalities_
>
> We thank you for this valuable suggestion. We fully agree that cross-modal interaction is important in LVLMs, especially in increasingly unified multimodal architectures.
>
> **For explicit cross-modal interaction**, we provide an initial analysis in Fig. 3 (bottom right) and Figs. 8 to 10, where the color of each visual patch reflects its cross-modal attention relevance during answer generation. Beyond these case studies, we further analyze cross-modal interaction from three aspects:
> 1. We study the redundancy of attention across **different modality pairs**, as shown in our response to **Reviewer QLh3 `W4 & Q4`**.
> 2. We examine the **task-dependent role of attention**, namely how sensitive different tasks are to SAP, as shown in our response to **Reviewer 7pD3 `W1 & Q3`**.
> 3. We construct **token-level visual attention trajectories**, showing how the model’s visual attention changes when generating specific output tokens. The details are provided in [https://anonymous.4open.science/r/ICML_rebuttal-anonmized], which is anonymous and compliant with the ICML author response policy.
>
> **For generalization beyond the visual modality**, we also extend RID and MixIG to the language modality and evaluate them on the Qwen-2.5 series. The results are as follows:
>
> |Model|Module|RID|MixIG|
> |-|-|-:|-:|
> |Qwen-2.5-1.5B|Attention|0.08|0.44|
> |Qwen-2.5-1.5B|FFN|0.22|0.07|
> |Qwen-2.5-3B|Attention|0.09|0.52|
> |Qwen-2.5-3B|FFN|0.30|0.11|
>
> These results show that RID and MixIG also identify a clear division of labor in pure language models: attention is associated with stronger mixing behavior, while FFN is associated with larger RID. This is consistent with our findings in LVLMs and suggests that the proposed metrics are not limited to LVLMs, but can also generalize to pure language models and potentially other modalities.
>
> ---
> > `Limitations`:  _Further analysis_
>
> We thank you for the constructive suggestions regarding the limitations of our work. We will incorporate these points into the limitation discussion in the revised paper. At this stage, we also provide additional preliminary analyses:
> 1. The effect on model logits across different tasks, as discussed in our response to **Reviewer 7pD3 `W1 & Q3`**.
> 2. How the proposed findings can be used to guide model training, as discussed in our response to **Reviewer QLh3 `W2 & Q2`**.

---

> > ### Author Rebuttal · Reviewer_qfjM · 2026-04-02
> >
> > Thanks for your detailed response. I decide to maintain my positive score.
> >
> > A core concern is that, if we differentiate image tokens and text tokens. It may make the model and inference system too complex for practical use. Therefore, it would be better if all tokens could be processed in a unified way. For example, replace self-attention for all tokens, rather than just image tokens.
> > This concern may be out of the scope of the paper. Therefore, I decided to maintain my positive score.

---

> > > ### Author Response · Authors · 2026-04-02
> > >
> > > We sincerely thank you for the positive assessment and for maintaining the score after reading our response.
> > >
> > > Our core contribution is to provide a **unified** evaluation framework that places FFN and attention in the same analytical space, and under this framework, to reveal the redundancy of visual attention through SAP-based intervention. Our intention is not to overturn the current unified modeling paradigm, but to provide evidence that can help improve the design of vision-language models within it.
> > >
> > > We also fully agree with Reviewer qfjM that a unified model is necessary in practice. In this sense, our motivation is not to advocate separating image and text processing into a more complex system, but to highlight where the current architecture may still admit optimization. As suggested by our preliminary results in the response to Reviewer QLh3 `W2 & Q2`, such optimization appears feasible at both inference and training time.
> > >
> > > Building on our findings, we would also like to share an additional perspective. In language models, recent studies have shown that it is possible to identify critical attention heads or token-level attention structures and then optimize or calibrate them for better performance [1][2][3]. Since current LVLM decoders are inherited from language models and visual tokens are aligned into the same decoder representation space through the projector, this suggests a unified optimization direction: under the same decoder-side representation, methods for identifying and optimizing critical attention structures in language models may be transferable to LVLMs. In this sense, **our framework can serve as a basis for locating which visual tokens or visual-attention interactions are truly critical**, and then optimizing them during training to improve the learning of unified representations. This is also the direction of our ongoing future work.
> > >
> > > We sincerely thank you again for your insightful comments. We will further analyze how our metrics can be used to improve unified representation learning and incorporate these findings into the revised version.
> > >
> > > **References**
> > > [1] Yin, K., et al. *Which Attention Heads Matter for In-Context Learning?* ICML 2025.
> > >
> > > [2] Yu, Z. et al. *Unveiling and Harnessing Hidden Attention Sinks: Enhancing Large Language Models without Training through Attention Calibration.* ICML 2024.
> > >
> > > [3] Gu, X., et al. *When Attention Sink Emerges in Language Models: An Empirical View.* ICLR 2025.

---

### Official Review · Reviewer_oL2V · 2026-03-09

**Soundness:** 2
**Presentation:** 1
**Significance:** 2
**Originality:** 2
**Overall Recommendation:** 4
**Confidence:** 3

**Summary:**

This project mainly offers two measurement for LVLM: **Representation Information Discrepancy (RID)** for quantifying the representation dynamic and **Mixing Information Gain (MixIG)** for information dynmaics at different hidden states. By looking at the proposed metrics and designing a series of empirical studies, the authors are able to assign different functional role to the attention heads andn MLPs inside transformer decoder, which provides an interpretability perspective.

**Compliance With Llm Reviewing Policy:**

Affirmed.

**Final Justification:**

In my opinion, this submission will need major revision including updates on the narratives of manifold learning, on the distinction from previous studies, and on more profound impacts. That says this submission is marginally above the acceptance threshold, but I would not mind if paper is rejected.

**Key Questions For Authors:**

Please see my questions in the weakness part. I am more than willing to adjust my score if the authors can solve my concerns actively and effectively.

**Limitations:**

yes

**Strengths And Weaknesses:**

## **Soundness**
### **Strengths**
All proposed metrics have theoretical analysis and corresponding experimental results.

### **Weakness**
1. RID: I don't see a proper reason for adding up the spectrum change and the support innovation as a single metric. These two terms $\Delta\mathcal{S}$ and $\Delta\mathcal{D}$ clearly stand for different mathematical meanings.

2. Experimental design: I am sure that adding or replacing the meaning of stochastic baseline, especially adding or replacing with Gaussian noise, will make the model behavior highly unpredictable and irrational. Also, for observation 3, how does figure 3 resonate with the conclusion in lines 363-375, right-hand side?

## **Presentation**
### **Weakness**
Honestly, I don't appreciate the narrative style of connecting to fancy mathematical terms, especially since the main contribution can only rely on basic linear algebra. For example, the spectrum change in RID relies on rank information, and there seems to be no necessity to connect to the manifold learning. Also, the support for innovation is just how the change in the token space and feature space, the authors fail to provide a good motivation for considering both representation spaces. In all, I don't see a reason to decorate the paper with manifold learning.

For experiments, as I point out in the soundness part, the main context lacks sufficient explanation and discussion of design choice and baseline selection. Also, in the first 7 pages, the authors focus on elaborating the proposal metrics and their applications. However, the last experiment with *shared attention prior* seems to be a different approach and independent of the previously delineated findings.

## **Significance**
### **Weakness**
Even though the paper uses new terminology to describe the functional difference between attention modules and MLPs, the conclusion does not provide further distinction from previous related studies as cited in the paper. Also, the proposed measurements act on the transformer decoder, which is not necessarily studied or held on LVLM only. Failing to differentiate this project from previous studies diminishes the main contribution.

## **Originality**
### **Strengths**
Although I do not find much excitement about the proposed metrics, the intention to rely on interpretability in mathematics is meaningful.

### **Weakness**
Nevertheless, there are still a number of relevant references that the authors miss in their citation. Those works also focus on or are tangent to the functional difference of attention modules and MLPs inside LLMs. I would suggest that the authors check the following papers and try to compare the proposed findings with their conclusions.

[1] Basu, Samyadeep, et al. "Understanding information storage and transfer in multi-modal large language models." Advances in Neural Information Processing Systems 37 (2024): 7400-7426.

[2] Meng, Kevin, et al. "Mass-editing memory in a transformer." arXiv preprint arXiv:2210.07229 (2022).

[3] Kobayashi, Goro, et al. "Analyzing feed-forward blocks in transformers through the lens of attention maps." arXiv preprint arXiv:2302.00456 (2023).

[4] Yin, Kayo, and Jacob Steinhardt. "Which attention heads matter for in-context learning?" arXiv preprint arXiv:2502.14010 (2025).

[5] Meng, Kevin, et al. "Locating and editing factual associations in GPT." Advances in neural information processing systems 35 (2022): 17359-17372.

[6] Yao, Yunzhi, et al. "Knowledge circuits in pretrained transformers." Advances in neural information processing systems 37 (2024): 118571-118602.

---

> ### Author Rebuttal · Authors · 2026-03-30
>
> We sincerely thank you for your thoughtful review. In response to your comments, we have carefully prepared a point-by-point reply and will incorporate the corresponding revisions into the final version.
>
> ---
> > `On presentation`
>
> We apologize for the confusion caused by our presentation. In the final version, we will adjust the paper to emphasize geometric and information-theoretic intuition, and use subspace-preserving/expanding as the main narrative throughout, including corresponding revisions to the abstract, method, and experiment sections.
>
> The key assumption of our work is that hidden representations learned by the model are low-rank, and the most direct conceptual support for this view comes from the manifold hypothesis, i.e., the idea that high-dimensional representations often concentrate near lower-dimensional structure. Therefore, our discussion of manifold is only intended to provide intuition for why a low-dimensional geometric description is natural in this setting.
>
> To address your concern, we will move these discussions to the **appendix** and present manifold only as **optional intuition and theorem-level background**, rather than as part of the main technical narrative.
>
> ---
> > `On significance and originality`
>
> We agree that clearly distinguishing our work from prior studies is essential. We will revise the related work accordingly and include a more explicit discussion of [1]-[6].
>
> First, our work addresses a different question from [1]-[6]. Those studies mainly analyze module importance, attention targets, or information pathways. Their core questions are typically: which module matters, and where does information flow? This perspective is highly valuable for knowledge editing, circuit analysis, and module attribution. By contrast, our work focuses on a different level of analysis: **how a module changes the representation space**. The goal of RID and MixIG is not only to identify whether a module is important, but to characterize its update on the shared residual stream.
>
> Second, compared with prior work cited in our paper, our contribution is to place attention and FFN in the same analytical coordinate system, enabling a direct comparison between them rather than separate analyses of attention maps, FFN memories, or head-level causal effects. Therefore, our conclusion is not obtained by juxtaposing different analyses, but by a direct comparison within a unified metric space.
>
> Third, we agree that significance should go beyond descriptive analysis. The practical value of the proposed metrics is that they can guide both inference-time intervention and training-time design, as further supported by our response to **Reviewer QLh3 `W2 & Q2`**. We will make this practical significance more explicit in the revised version.
>
> ---
> > `On RID`
>
> We understand the concern that $\Delta S$ and $\Delta D$ capture different mathematical notions. In RID, external information injection can manifest through two complementary channels: changes in effective dimensionality, measured by $\Delta S$, and novelty in subspace support, measured by $\Delta D$.
>
> From the perspective of completeness, both terms are necessary. Using either term alone may miss important cases, such as subspace change with little spectral variation. For this reason, we aggregate the two normalized components as a total innovation budget. The normalization removes scale mismatch and places both terms in a comparable range. As shown in Appendix E.1, the resulting scalar RID is bounded in $[0,2]$.
>
> To make this clearer, we will revise the manuscript accordingly. In the method section, we will present $\Delta S$ and $\Delta D$ explicitly as a **two-dimensional quantity**, i.e., $(\Delta S,\Delta D)$, and clarify that the scalar RID is only their aggregated summary score.
>
> ---
> > `On Experiments`
>
> Our purpose in introducing Noise $\Delta$ and Noise QKV in Sec. 4.2 is to use them as **negative controls** for validating whether RID and MixIG can distinguish unstructured perturbations, rather than to explain the mechanism of real model reasoning.
>
> Moreover, the noise is not injected arbitrarily. We match the noise distribution to the distribution of real attention updates, and Appendix Theorem E.3 further discusses expectation consistency under noise injection, in order to avoid trivial differences caused by mean shift.
>
> To further demonstrate the stability of this conclusion, we repeat the experiment in Table 1 with 10 random seeds and report the mean $\pm$ std below:
>
> |Module|RID|MixIG|
> |-|-:|-:|
> |Noise $\Delta$|$0.61\pm0.023$|$-0.80\pm0.061$|
> |Noise QKV|$0.44\pm0.023$|$-0.50\pm0.064$|
> |Attention|$0.06\pm0$|$0.61\pm0$|
> |FFN|$0.21\pm0$|$0.02\pm0$|
>
> These results are stable across random seeds. We will include this table in the final version of the paper.
>
> We apologize for the confusion caused by Obs. 3 and Fig. 3. In the revision, we will separate the RID/MixIG curves from the patch interaction graph to make this connection clearer.

---

> > ### Author Rebuttal · Reviewer_oL2V · 2026-04-02
> >
> > Thank you for the authors' active and detailed rebuttal comment, not only to me but to all reviewers. I do see many interesting discussions; however, I would still feel less pleased with some answers to the core of my concerns.
> >
> > The paper is shaped to discuss the functional difference between attention modules and MLPs, in which I do consider that this can be done on LLMs solely. While for more significant concerns, I still do not find much difference from the conclusion of previous work studying on this topic. As the authors mentioned in lines 029-035 righthand side: *attention retrieves and FFNs memorizes*. How the proposed claim *"attention acts as a manifold-preserving operator; FFNs serve as manifold-expanding operators"* differing from the previous finding is still remain unspecified.
> >
> > Moreover, if the authors agree to *present manifold only as optional intuition and theorem-level background*, can the author explain the idea of the terms "innovation" and "reconfiguration"? Are they not synonymic to 'memorization" and "retrieval" in the context of LLM's intrinsic mechanism?
> >
> > Considering the amount of active and effective efforts in addressing reviewers' concerns, I will raise my assessment to **3** at this point.

---

> > > ### Author Response · Authors · 2026-04-04
> > >
> > > We sincerely thank you for raising our score. We understand your concern as follows:
> > >
> > > > Are "innovation" and "reconfiguration" merely a mathematical restatement of the familiar narrative that attention retrieves and FFN memorizes, without yielding genuinely new conclusions or actionable insights?
> > >
> > > We address this concern as follows.
> > >
> > > ## 1. Why do we study LVLMs?
> > > Although our metrics apply to any LLM, as discussed in our response to Reviewer qfjM `W2`, we focus on LVLMs because visual tokens typically outnumber text tokens by $30\times$ in practice, making their cost dominant under the quadratic $O(N^2)$ attention complexity, while our SAP results show that a substantial portion of visual attention can be replaced by Gaussian noise without hurting performance; this mismatch between computation and information is what motivates our study of LVLMs.
> > >
> > > ## 2. What is the core difference from prior work?
> > > ### **2.a.** Difference in method
> > > Prior work on module functionality is based on statistical attribution and tracing. Our work instead uses **information-theoretic and geometric analysis**.
> > >
> > > These two lines answer different questions. Attribution-based methods can localize what a neuron (or circuit) stores. By contrast, our framework asks how a module transforms the shared residual stream. This lets us detect phenomena such as weak mixing or insufficient innovation even when they are not tied to a specific fact or circuit. For example, MixIG identifies weak interaction in middle layers, and this diagnosis directly leads to effective intervention and improved fine-tuning behavior.
> > >
> > > ### **2.b.** Difference in perspective
> > > A second difference lies in the perspective. Much of the existing literature is strongest at identifying **what function is present** in the model. Our framework is designed to diagnose **what is insufficient** in the residual update itself.
> > >
> > > RID asks whether a module injects genuinely new structure into the representation and MixIG asks whether a module meaningfully mixes token information. In this sense, “innovation” and “reconfiguration” are not synonyms of “memory” and “retrieval.” They are representation-level properties of the update itself. This deficiency-oriented view is also what makes the framework actionable: once a layer is identified as weak in mixing or low in contribution, it can be directly tested and optimized.
> > >
> > > ### **2.c.** Difference in conclusions and impact
> > > For FFNs, prior memory-based work explains how FFN parameters encode and recall patterns or facts. Our question is different: regardless of whether information comes from parameters or context, **what does the residual update do to the representation geometry**? This naturally goes beyond factual memory.
> > >
> > > A preliminary example is a **model-level layer-wise RID** analysis. We compare Qwen-2.5-VL with two RL-enhanced reasoning models layer by layer, where the percentages denote the RID between corresponding layers in the two models:
> > >
> > > |base-model|rl-model|0|25%|50%|75%|100%|
> > > |-|-|-:|-:|-:|-:|-:|
> > > |qwen|Ocean-r1|.05|.07|.17|.24|.12|
> > > |qwen|mm-eureka|.03|.04|.09|.12|.05|
> > >
> > > The main RID increase appears in the middle-to-late layers, suggesting that reasoning capability is injected there. We then replace these layers in Ocean-r1 with the corresponding base-model layers, yielding a **mixed-base model**, and conversely replace the corresponding layers in the base model with those from Ocean-r1, yielding a **mixed-reason model**. and evaluate how many samples follow the expected output format (contain the `<think>` and `<answer>` tags):
> > >
> > > ||base|mixed-base|mixed-reason|reasoning|
> > > |-|-:|-:|-:|-:|
> > > |think-prompt|22|27|57|100|
> > >
> > > Here, "think-prompt" triggers the standard thinking mode described in *Appendix B.2*. Under "think-prompt", the mixed-base largely loses the expected reasoning behavior after these layers are replaced. This suggests that RID can help localize the injection of a higher-order capability, rather than only factual memory, which also reflects a **key difference** between our work and prior studies.
> > >
> > > For attention, prior work on attention circuits explains what algorithms attention can implement. Our claim is different: in the **visual side of current LVLM decoders**, many attention layers exhibit weak mixing and low contribution, and their scoring is replaceable under intervention. Therefore, even if strong attention circuits exist in principle, current LVLMs do not consistently convert expensive visual attention scoring into necessary output discrimination. This is a new finding from our SAP experiments.
> > >
> > > ---
> > >
> > > We sincerely thank you for further raising your score. As promised, **we have updated the narrative around manifold, and revised the related work, discussion, and appendix** to better clarify the distinctions from prior studies as well as the broader applicability of our framework. Your constructive suggestions helped make the paper much clearer. **These revisions do not affect our ` core method or conclusions `**.

---

### Official Review · Reviewer_QLh3 · 2026-03-10

**Soundness:** 3
**Presentation:** 3
**Significance:** 2
**Originality:** 3
**Overall Recommendation:** 4
**Confidence:** 3

**Summary:**

This paper studies the internal mechanisms of Large Vision-Language Models (LVLMs) through a geometry-based analytical framework. The authors introduce two metrics—Representation Information Discrepancy (RID) to measure semantic innovation and Mixing Information Gain (MixIG) to quantify information reconfiguration within the representation manifold. Using these metrics to analyze 15 LVLM families, the study reveals a functional decoupling between Transformer components: FFN layers primarily introduce new semantic directions, while attention modules mainly redistribute existing information.

A key finding is that visual attention in current LVLMs is often redundant or inefficient. The authors show that replacing learned visual attention weights with simple non-parametric priors or even Gaussian noise in many layers causes little or no performance degradation and can sometimes improve accuracy on benchmarks such as POPE, MMMU, and MathVista.

Overall, the work provides a theoretical framework for interpreting Transformer updates and highlights potential inefficiencies in current vision-language attention mechanisms.

**Compliance With Llm Reviewing Policy:**

Affirmed.

**Final Justification:**

After considering both the paper and the authors’ rebuttal, I am now more supportive of this submission because it offers an original and potentially impactful framework for analyzing LVLM internals, the rebuttal substantially improves the reliability and practical value of the main claims, and although the deeper causal account of attention redundancy remains incomplete, my principal concerns have been sufficiently addressed to change my assessment in a positive direction.

**Key Questions For Authors:**

1.On the Principle for Automated Layer Selection: In your intervention experiments, you apply the Shared Attention Prior (SAP) to a specific range of layers (e.g., layers 1–27). While the results are compelling, the selection of this range seems somewhat empirical or based on post-hoc observation. Is there a principled, automated way to use the RID and MixIG metrics to dynamically identify these "redundant" layers for a completely unseen model architecture? If the authors can provide a metric-driven threshold for layer selection, it moves the paper from being an interesting observation to providing a robust, actionable tool for architectural optimization.

2.Training vs. Inference Redundancy: The paper brilliantly demonstrates that current LVLMs "get lost" in attention during inference. However, is this redundancy a result of the training objective (i.e., the model eventually learns to rely on FFNs for semantic innovation) or an architectural bottleneck (i.e., the attention mechanism is inherently inefficient for high-dimensional visual tokens from the start)? Have you conducted any preliminary tests to see if these models could be trained from scratch with fixed SAP layers? Answering this would clarify whether the paper is identifying a flaw in how we train these models or a fundamental flaw in the Transformer architecture for multi-modal tasks. Training-time efficiency would vastly increase the paper’s impact.

3.Computational Scalability of RID and MixIG: The calculation of your metrics involves Singular Value Decomposition (SVD) and entropy estimation on high-dimensional hidden states. As LVLMs move toward handling long-context sequences (thousands of tokens), what is the computational overhead of these metrics? Can they be efficiently approximated for real-time model diagnostics? This addresses the "practicality" of the work. If the diagnostic tool itself is too computationally expensive to run on large-scale models, its adoption as a standard evaluation framework might be limited.

4.Distinguishing Intra-modal and Cross-modal Redundancy: When you replace the attention weights with SAP, you are essentially simplifying the interaction between all tokens. Does your analysis allow for a distinction between visual-visual self-attention redundancy and visual-text cross-modal attention redundancy? Which of these is the primary driver of the "lost in attention" phenomenon? This would deepen the mechanical understanding of the work. If the redundancy is purely in how visual tokens talk to each other, the solution is different than if the bottleneck lies in how vision and language are fused. Clarifying this would significantly strengthen the "Scientific Discovery" aspect of the paper.

**Limitations:**

No

Refine the Scope of "Redundancy": The authors should explicitly caution that their findings of "attention redundancy" might be specific to the current architectural paradigms of LVLMs. It would be beneficial to discuss whether this redundancy is a permanent trait of multi-modal fusion or a byproduct of sub-optimal training objectives (e.g., standard next-token prediction).

Generalizability across Domains: The limitation section should clarify if these findings hold for specialized domains like medical imaging or high-resolution satellite analysis, where fine-grained visual attention might be more critical than in general-purpose benchmarks.

Security and Robustness Implications: Regarding negative social impacts, the authors could discuss whether replacing learned attention with simple priors (like SAP) makes the model more or less susceptible to adversarial attacks or "jailbreaking" through visual prompts. A more "rigid" attention mechanism might inadvertently introduce new security vulnerabilities.

Hardware Bias: The discussion could address how these theoretical metrics (RID/MixIG) translate to actual hardware efficiency. While they point toward redundancy, the practical realization of "faster" models often depends on hardware-specific kernels, which the paper does not deeply explore.

**Strengths And Weaknesses:**

**Strengths**
The paper presents a technically rigorous framework for analyzing the internal mechanisms of Large Vision-Language Models (LVLMs). By grounding the analysis in the manifold hypothesis and information theory, the authors introduce two metrics—RID and MixIG—to characterize representation updates in terms of semantic innovation and information reconfiguration. This formulation provides a clear perspective on the roles of Attention and FFN modules within the Transformer architecture. The use of stochastic baselines to validate the proposed metrics further strengthens the methodological rigor.

The empirical findings are particularly interesting. The analysis reveals a functional decoupling where FFNs mainly drive semantic innovation while attention primarily redistributes existing information. Moreover, the observation that simple attention priors can replace learned visual attention with little performance loss suggests potential architectural inefficiencies in current LVLMs.

The experimental evaluation is extensive, covering 15 LVLM families, which increases the credibility and generality of the conclusions. The paper is also well organized and clearly written.

**Weaknesses**
A key concern is the heuristic selection of layers for SAP/noise interventions. While the middle layers are shown to be often redundant, the criteria for choosing exact ranges (e.g., layers 1–27) are manual, and an automated, metric-driven approach would strengthen the analysis.

The study primarily focuses on post-hoc analysis and inference-time interventions, leaving open whether these insights can guide training strategies, such as pretraining with fixed SAP layers to save compute.

The computation of RID and MixIG relies on SVD and entropy estimation on high-dimensional hidden states, but the paper does not discuss the cost or scalability of these metrics for long-context LVLMs.

Finally, while the functional decoupling of FFNs and Attention is well documented, the paper provides limited insight into why this occurs—whether due to limitations of visual encoders or architectural constraints. A deeper discussion of underlying causes would enhance the interpretive contribution.

---

> ### Author Rebuttal · Authors · 2026-03-30
>
> Thank you for the thoughtful review and for highlighting both the strengths and the open questions of our work. In the following, we address each point in turn.
>
> ---
> > `W1 & Q1`: _MixIG-guided attention intervention_
>
> We understand your concern that if the intervention experiment is purely empirical, it would weaken the significance of the method. To address this, we design a MixIG-guided attention intervention. Specifically, we first use a z-score criterion on the layer-wise MixIG distribution to automatically identify weak-interaction layers; on Qwen-2.5-VL-3B, this criterion selects about 50\% of the layers, on which we then apply SAP (**MixIG-SAP**). The corresponding results are reported in our response to **Reviewer qfjM `W1`**.
>
> To show that this criterion can transfer to other models, we report the following results on Qwen-2.5-VL-7B:
>
> |Model|General|Reasoning|OCR|Doc\&Chart|
> |-|-:|-:|-:|-:|
> |Qwen-2.5-VL-7B|84.5|70.2|94.9|72.2|
> |+ fixed-SAP|85.7|71.1|93.5|73.0|
> |+ MixIG-SAP|86.1|71.3|95.1|73.1|
>
> These results show that the z-score-based MixIG-SAP remains stable and further improves over fixed-SAP on an unseen model.
>
> ---
> > `W2 & Q2`: _Training vs. Inference Redundancy_
>
> We thank you for recognizing our analysis of the reasoning process. We have conducted preliminary training experiments on ScienceQA using a Qwen-2.5-VL-based VLM implemented in LLaMA-Factory. Under the same LoRA setup, we compare the following three settings:
>
> 1. Standard LoRA.
> 2. LoRA with a randomly selected 50% of attention layers frozen.
> 3. LoRA with the 50% of attention layers identified as redundant by MixIG-SAP frozen.
>
> Each setting is trained for 10,000 steps, and we report the number of trainable parameters, training time per step (sec/step) and test accuracy below.
>
> |Method|Trainable Params.|Time per step|Acc.|
> |-|-:|-:|-:|
> |Vanilla|9.23M|2.91s|91.6|
> |Random|4.62M|2.12s|87.9|
> |MixIG-SAP|4.62M|2.12s|91.6|
>
> These results show that MixIG-SAP uses fewer trainable parameters and less training time, while achieving higher accuracy.
>
> ---
> > `W3 & Q3`: _computation cost_
>
> We thank you for raising this important practical concern. We agree that scalability is critical if these metrics are to be useful beyond small-scale analysis.
>
> As you pointed out, the main cost of our metrics comes from two parts: **SVD for RID** and **token-wise similarity for MixIG**. This cost can be reduced in the long-sequence regime, since hidden representations are often approximately low-rank in practice.
>
> Specifically, randomized SVD can be used to estimate the leading $k$ singular directions for RID with complexity $O(nd\log k+(n+d)k^2),$ and MixIG can then be approximated in the same low-rank token space with complexity $O(n^2 k),$ where $n$, $d$, and $k$ denote the number of tokens, hidden dimension, and effective singular directions, respectively.
>
> Accordingly, the total cost of approximating RID and MixIG for one layer is $O(nd\log k + (n+d)k^2 + n^2 k).$ For comparison, the per-layer forward complexity of a standard Transformer block is $O(n^2 d + n d^2).$ Therefore, the ratio to one layer forward pass is
> $$\frac{O(nd\log k+(n+d)k^2+n^2k)}{O(n^2d+nd^2)}.$$
> Under long-context settings with large $n$ and moderate effective rank $k$, this ratio becomes very small. In our experiments on high-resolution inputs with thousands of visual tokens, the approximate **computation costs only about $1/60$ of one inference pass** on average over 100 samples, while keeping the **approximation error within 5\%**.
>
> ---
> > `W4 & Q4`: _Redundancy attribution_
>
> Our conclusion is that attention redundancy exists **both** within the visual modality and in vision-text interaction. To verify this, we use a standard attribution-style intervention based on the SAP setup in Section 4.3. Specifically, we replace the attention scores for different modality pairs with Gaussian noise, and then examine the output logits of Qwen-2.5-VL-3B. Here, the logits are measured as the difference between the logit of the correct answer and the mean logit of the other answer options during prediction, which reflects the model’s confidence in selecting the correct answer.
>
> |Intervention|General|Reasoning|OCR|Doc\&Chart|
> |-|-:|-:|-:|-:|
> |output to vision|3.95|2.12|5.76|3.42|
> |vision to vision|3.91|2.09|5.54|3.36|
> |output to question|1.12|0.38|1.21|0.63|
>
> These results show that perturbing attention involving the visual modality, especially output-to-vision attention, has much **weaker impact** on the model confidence than perturbing text-side attention (output-to-question). In other words, after visual features are projected into the language space, the decoder does not rely on visual-side attention as effectively as it relies on text-side interaction. This provides further evidence that the decoder’s language-centered initialization is a plausible source of the identified redundancy in both intra-visual interaction and cross-modal fusion.

---

> > ### Author Rebuttal · Reviewer_QLh3 · 2026-04-04
> >
> > After considering the authors’ rebuttal, I am inclined to revise my assessment upward, as the paper presents a genuinely interesting and potentially useful framework for analyzing module-level functionality and attention redundancy in LVLMs.

---

> > > ### Author Response · Authors · 2026-04-04
> > >
> > > Thank you for your time and effort throughout the review process, and for acknowledging our rebuttal. We are very glad that our response effectively addressed your questions.

---

### Official Review · Reviewer_7pD3 · 2026-03-12

**Soundness:** 4
**Presentation:** 3
**Significance:** 3
**Originality:** 4
**Overall Recommendation:** 5
**Confidence:** 2

**Summary:**

This paper proposes an information-theoretic and geometric framework for analyzing residual-stream updates in large vision-language models. Using the proposed RID and MixIG metrics, it argues that attention mainly performs reconfiguration while FFNs mainly provide innovation, and further claims that decoder visual attention is often redundant enough that replacing learned attention scores with simple priors or even noise does not hurt performance.

**Compliance With Llm Reviewing Policy:**

Affirmed.

**Final Justification:**

This paper analyzes the residual-stream updates in Large Vision-Language Models, suggesting that decoder visual attention exhibits significant redundancy. In my initial review, I questioned the generalizability of the Shared Attention Prior (SAP) experiments and requested further causal analysis to explain performance variations in spatially sensitive tasks like OCR.

The authors' rebuttal adequately addressed my empirical concerns. The inclusion of a uniform SAP configuration across architectures and the new MixIG-SAP variant demonstrated that the attention redundancy is not merely an artifact of model-specific tuning. Additionally, the new logit margin analysis provided a reasonable explanation for task-specific sensitivities.

After reading the comments from the other reviewers, I realized there may be theoretical or technical aspects of the submission that I did not fully grasp, which is why I have decided to lower my confidence score. However, because the authors resolved the specific empirical issues I raised in my own review, I am maintaining my initial positive score.

**Key Questions For Authors:**

1. Which parts of the “get lost in attention” claim are directly supported by the intervention results, and which parts remain interpretive extrapolation?
2. nstead of the model-specific tuning described in the Appendix, would the SAP results remain robust if a uniform layer range (e.g., the top 20%) and consistent head selection criteria were applied across all models?
3. As shown in Appendix Table 5, performance fluctuations are more pronounced in OCR and document/chart understanding tasks. What is your rationale for why the role of attention becomes increasingly critical in these complex reasoning tasks compared to simpler ones?

**Limitations:**

yes

**Strengths And Weaknesses:**

**Strength**

The paper tries to move beyond purely attribution-based analysis by proposing a more principled module-level framework. The empirical section is large in scope, and the effort to validate the metrics through noise baselines and cross-model consistency is a genuine strength. The intervention results in which predefined attention priors remain competitive is also provocative and likely to generate discussion.

**Weakness**
- More causal analysis is needed to explain why substituting learned attention with Gaussian noise leads to sustained or superior performance.
- In the SAP experiments, the layers and heads selected for intervention vary across different models. This raises concerns that the findings may not represent a universal phenomenon, but rather a post-hoc selection of successful configurations through extensive ablation.

---

> ### Author Rebuttal · Authors · 2026-03-30
>
> We sincerely thank you for your careful comments and thorough understanding of our paper! Here we give point-by-point responses to your comments and describe the revisions we made to address them.
>
> ---
>  > `W2 & Q2`: _Generalization and robustness of SAP_
>
> We thank you for this valuable question. We understand the concern that the gains might arise from model-specific tuning rather than a general phenomenon. To directly address this point, we conduct two progressively stronger analyses.
>
> First, we apply a **uniform intervention** setting across models by fixing the affected layers (top 25\%-50\%) and heads (0.3-0.6), and then replacing the corresponding attention scores with noise. In each entry below, the numbers before and after “/” denote the original result and the SAP result, respectively.
>
> |Model|General|Reasoning|Math|Doc&Chart|OCR|
> |-|-:|-:|-:|-:|-:|
> |Qwen-2.5-VL-7B|84.5/85.7|70.2/71.1|43.7/44.2|72.2/73.0| 94.9/93.5|
> |LLaVA-1.5-7B|65.9/69.0|49.6/49.8|25.3/27.1|33.8/34.2|64.7/62.6|
> |LLaVA-OV-7B|80.8/81.0 |64.8/65.6|39.0/41.1|57.4/58.3|89.3/88.7|
>
> These results show that the SAP effect remains observable under a shared intervention protocol across different architectures, which suggests that our findings are not solely an artifact of model-specific intervention choices.
>
> Second, beyond the fixed SAP setting in the appendix, we also evaluate a MixIG-based SAP variant (**MixIG-SAP**). Specifically, we identify the 50\% of layers with the smallest MixIG contributions and then apply SAP to them. The results are reported in our response to **Reviewer \#qfjM `W1`**. This provides an additional, principled selection criterion and further supports the robustness of the SAP phenomenon beyond fixed configuration choices.
>
> ---
> > `W1 & Q3`: _More causal analysis and task-dependent role of attention_
>
> We thank you for this important question. To directly address this point, we conducted an additional intervention study on attention allocation. Specifically, we intervene on attention allocation and measure the change in
> $$\delta=\mathrm{logits}(\mathrm{ground\\_truth})-\text{avg}(\text{logits}(\mathrm{other\\_options})),$$
>
> where a larger $\delta$ indicates stronger discrimination between the correct answer and the other options. We compare three settings on the datasets used in Tables 4-7:
> - The original model
> - **Random-SAP**: randomly selects 50\% of the layers and applies SAP to them
> - **MixIG-SAP**: applies SAP to 50\% of the layers with the smallest MixIG, as described in our response to `W2 & Q2`
>
> |Model|General|Reasoning|OCR|Doc&Chart|
> |-|-:|-:|-:|-:|
> |Qwen-2.5-VL-3B|3.86|2.05|5.54|3.37|
> |+ Random-SAP|3.88|2.06|5.24|3.21|
> |+ MixIG-SAP |3.98|2.12|5.76|3.48|
>
> These results support two observations. First, replacing learned attention with random patterns does not consistently reduce, and may even improve, the logit margin, suggesting that part of the learned attention is **redundant** for final answer discrimination. Second, the improvement in $\delta$ is especially clear on OCR, where MixIG-SAP gives the largest gain.
>
> Our interpretation for why attention becomes more critical in OCR and document/chart tasks has two parts. First, OCR and document/chart tasks rely **more heavily** on fine-grained spatial grounding and precise local correspondence. In Table 5, however, we use fixed-SAP, whose uniform intervention may **perturb** layers that still carry necessary information interaction. As a result, these tasks are more sensitive to such intervention and show larger fluctuations. Second, under MixIG-SAP, the intervention is applied to layers with **weak** interaction. In this case, replacing the attention scores with noise does not disrupt critical alignment, and may even promote information exchange across patches, leading to more stable or improved performance.
>
> ---
> > `Q1`: _"Get lost in attention"_
>
> We thank you for this helpful question. In our paper, we present four main observations. Obs. 1 and Obs. 2 are mainly about module-level interpretability, while **Obs. 3 and Obs. 4** support the "get lost in attention" claim.
>
> More specifically, Obs. 3 is an interpretive inference based on our explanatory metrics, i.e., MixIG and RID. It suggests that attention often fails to allocate its capacity to the most relevant information, which motivates the "lost" characterization. However, this part is still diagnostic rather than directly causal.
>
> By contrast, Obs. 4 is directly supported by the SAP intervention results. In particular, replacing learned attention scores with predefined patterns, including Gaussian noise, does not hurt performance and can even improve it. This provides direct evidence that a substantial portion of learned attention scoring is replaceable.
>
> Therefore, our intended claim is progressive: Obs. 3 provides the interpretive basis for why attention may be “lost,” and Obs. 4 provides direct intervention-based support that this misallocated attention is not necessary for maintaining performance.

---

> > ### Author Rebuttal · Reviewer_7pD3 · 2026-04-04
> >
> > The authors have adequately addressed my main concerns. The uniform SAP setting and MixIG-SAP experiments effectively demonstrate the robustness and generalizability of the findings beyond model-specific tuning. The progressive distinction between diagnostic (Obs 3) and interventional (Obs 4) evidence also clarifies the scope of the claims. I maintain my positive score.

---

> > > ### Author Response · Authors · 2026-04-04
> > >
> > > We sincerely thank you for the time and effort devoted to reading our paper and rebuttal! We are very glad that we were able to address your concerns regarding the robustness of SAP and the scope of our claims.

---

### Decision · Program_Chairs · 2026-04-30

**Decision:**

Accept (regular)

**Comment:**

This paper presents an information‑theoretic and geometric framework for analyzing residual‑stream updates in LVLMs and offers a unified perspective on the distinct functional roles of attention and FFN modules. Reviewers agreed that the proposed RID and MixIG metrics provide a principled basis for module‑level analysis beyond attribution‑based methods, and that the SAP intervention results offer compelling empirical evidence of redundancy in current architectures.
The author rebuttal effectively addressed key concerns regarding SAP generalizability, token‑scope alignment, and scalability of the proposed metrics. While a deeper causal account of attention redundancy remains an open direction, the committee finds that the framework offers meaningful diagnostic insight into LVLM internals with potential implications for both inference‑time optimization and training‑time design.
Final Scores: 5 / 4 / 4 / 4
AC Decision: Agree with reviewer consensus (Accept)